# LLMDFA: Analyzing Dataflow in Code with Large Language Models

**Chengpeng Wang**[1], **Wuqi Zhang**[2], **Zian Su**[1], **Xiangzhe Xu**[1], **Xiaoheng Xie**[3], **Xiangyu Zhang**[1]

[1] Purdue University, [2] Hong Kong University of Science and Technology, [3] Ant Group

```
{wang6590, su284, xu1415, xyzhang}@purdue.edu
wuqi.zhang@connect.ust.hk, xiexie@antgroup.com
```

## Abstract

Dataflow analysis is a fundamental code analysis technique that identifies dependencies between program values. Traditional approaches typically necessitate successful compilation and expert customization, hindering their applicability and usability for analyzing uncompilable programs with evolving analysis needs in real-world scenarios. This paper presents LLMDFA, an LLM-powered compilation-free and customizable dataflow analysis framework. To address hallucinations for reliable results, we decompose the problem into several subtasks and introduce a series of novel strategies. Specifically, we leverage LLMs to synthesize code that outsources delicate reasoning to external expert tools, such as using a parsing library to extract program values of interest and invoking an automated theorem prover to validate path feasibility. Additionally, we adopt a few-shot chain-of-thought prompting to summarize dataflow facts in individual functions, aligning the LLMs with the program semantics of small code snippets to mitigate hallucinations. We evaluate LLMDFA on synthetic programs to detect three representative types of bugs and on real-world Android applications for customized bug detection. On average, LLMDFA achieves 87.10% precision and 80.77% recall, surpassing existing techniques with F1 score improvements of up to 0.35. We have open-sourced LLMDFA at `https://github.com/chengpeng-wang/LLMDFA`.

## 1 Introduction

Dataflow analysis is a formal method that identifies the dependence between values in a program [1]. Its primary objective is to determine whether the value of a variable defined at a particular line, referred to as a *source*, affects the value of another variable used at a subsequent line, referred to as a *sink*. This crucial information offers valuable insights into various downstream applications, such as program optimization [2] and bug detection [3, 4]. In Figure 1, for example, we can regard the variable $x$ at line 9 as a source and the divisors at lines 4, 11, and 14 as sinks and eventually detect a divide-by-zero

```
1  public class Demo {
2   public static int foo(int a, int b){
3     if (Math.abs(b) > 1)
4       System.out.print(a / b);//sink: b, safe
5     return b;
6   }
7   public static void main(String[] args){
8     int x;
9     x = Integer.parseInt(args[0]);//source: x
10    int y = x * x + 1;
11    int z = x / y; //sink: y, safe
12    z = x;
13    y = foo(y, z);
14    System.out.print(x / y);//sink: y, buggy
15  }
16 }
```

Figure 1: An example of DBZ bug

(DBZ) bug at line 14. The intuition is that $x$ at line 9 comes from user input and can be zero. If it can flow to the divisors, DBZ bugs may occur.

Despite decades of effort, current dataflow analysis techniques have drawbacks in terms of applicability and usability. First, many scenarios where dataflow analysis is needed involve incomplete and uncompilable programs, e.g., on-the-fly code flow analysis in Integrated Development

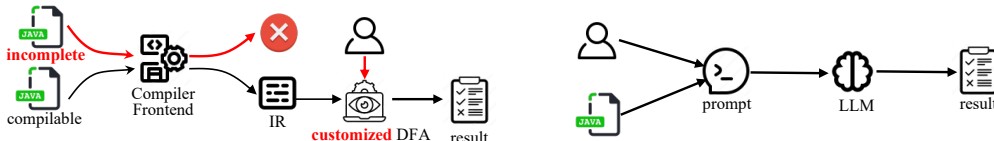

(a) Classical dataflow analysis relies on compilation and customization    (b) A new paradigm: LLM-powered dataflow analysis

Figure 2: Two different paradigms of dataflow analysis

Environments (IDEs). However, current techniques typically rely on intermediate representations (IRs) generated by compiler frontends, such as LLVM IR [5] produced by the Clang compiler, as shown in Figure 2(a). This reliance limits their applicability in analyzing incomplete programs, leading to the failure of analysis. Second, specific downstream tasks require customizing the analysis to fit specific needs, such as the detection of a specific bug type. In the DBZ detection, for example, the user has to extract variables potentially assigned with zero (sources) and variables used as the divisors (sinks). It is a challenging task for non-experts since they need a deep understanding of program representation (e.g., LLVM IR) for customization. This limitation hinders the usability of classical approaches to address evolving software analysis needs in real-world scenarios [6].

In the past year, there has been a vast proliferation of software engineering applications built upon large language models (LLMs), from which we observe the exceptional performance of LLMs in comprehending code snippets [7, 8, 9, 10]. Specifically, by treating LLMs as code interpreters and devising appropriate prompts, we can directly obtain semantic properties from source code. For instance, by constructing a prompt like "*Does the value of the variable $z$ used at line 13 depend on the value of variable $x$ defined at line 9*?", we can ascertain the dataflow fact between the two program values. Likewise, we can leverage LLMs to automate the extraction of specific sources and sinks by describing their characteristics as prompts. This empowers developers to tailor dataflow analysis to their specific requirements. As shown in Figure 2(b), such LLM-powered dataflow analysis gets rid of compilation and avoids complicated customization. In the rest of this paper, our demonstration is consistently within the context of a downstream task, and our references to sources and sinks specifically pertain to that task.

However, instantiating dataflow analysis using LLMs is far from trivial, as their hallucinations [11, 12, 13] threaten the reliability of results. First, the misidentification of sources and sinks leads to dataflow facts that are irrelevant to the user's interest, directly resulting in incorrect results of dataflow analysis. Second, sources and sinks can be distributed across multiple functions, entailing analyzing a large body of code that likely exceeds the input context limit. In addition, incorrect dataflow facts in single functions can accumulate and magnify, thereby impacting the overall performance. Third, the validity of a dataflow fact depends on the feasibility of the program path inducing the dataflow fact. If the path condition is deemed unsatisfiable, no concrete execution will occur along that path [4]. Regrettably, deciding the satisfiability of a logical constraint is a complex reasoning task that LLMs cannot effectively solve [14]. In Figure 1, for example, `gpt-3.5-turbo-0125` reports a DBZ bug at line 4 as a false positive because it cannot discover the unsatisfiable branch condition at line 3.

This paper presents LLMDFA, an LLM-powered compilation-free and customizable dataflow analysis. To mitigate hallucination, we decompose the analysis into three sub-problems, namely source/sink extraction, dataflow summarization, and path feasibility validation, which target more manageable tasks or smaller-sized programs. Technically, we introduce two innovative designs to solve the three sub-problems. First, instead of directly prompting LLMs, we leverage LLMs as code synthesizers to delegate the analysis to external expert tools like parsing libraries and SMT solvers [15], which effectively mitigates the hallucinations in the source/sink extraction and path feasibility validation. Second, we employ a few-shot chain-of-thought (CoT) prompting strategy [16] to make LLMs aligned with program semantics, which enables LLMDFA to overcome the hallucination in summarizing dataflow facts of single functions. Compared to traditional dataflow analysis, LLMDFA offers distinct advantages in terms of applicability and autonomy. It can be applied to incomplete programs, including those in the development phase. Additionally, it can autonomously create and utilize new tools with minimal human intervention, requiring no particular expertise in customizing dataflow analysis.

We evaluate LLMDFA in the context of bug detection. Specifically, we choose Divide-by-Zero (DBZ), Cross-Site-Scripting (XSS)[1], and OS Command Injection (OSCI)[2] in Juliet Test Suite [17]

---

[1]In an XSS bug, a dataflow fact across variables denoting different websites allows undesirable execution.

[2]Using an external input for OS command construction can lead to an OSCI bug.

for the evaluation. LLMDFA achieves high precision and recall when using different LLMs. For example, equipped with `gpt-3.5-turbo-0125`, it obtains 73.75%/100.0%/100.0% precision and 92.16%/92.31%/78.38% recall in the DBZ/XSS/OSCI detection. LLMDFA substantially outperforms a classic dataflow analyzer CodeFuseQuery [18] and an end-to-end solution based on few-shot CoT prompting, improving the average F1 score by 0.23 and 0.36, respectively. Besides, we evaluate LLMDFA upon real-world Android malware applications in TaintBench [19] and achieve 74.63% precision and 60.24% recall. It surpasses the two baselines with improvements in the F1 score of 0.12 and 0.35, respectively. To the best of our knowledge, LLMDFA is the first trial that leverages LLMs to achieve compilation-free and customizable dataflow analysis. It offers valuable insights into future works in analyzing programs using LLMs, such as program verification [20, 10] and repair [9].

## 2 Preliminaries and Problem Formulation

**Definition 1.** (**Control Flow Graph**) The control flow graph (CFG) of a given program $P$ is a labeled directed graph $G := (S, E_\ell)$. Here, $s \in S$ is a statement in the program. For any $(s, s') \in S \times S$, $E_\ell(s, s')$ is the boolean expression under which that $s'$ is executed just after $s$.

Figure 3 shows a (partial) CFG of the program in Figure 1. It depicts several important program facts, such as individual branch conditions and caller-callee relation. To simplify formulation, we introduce $V_{par}^f$, $V_{ret}^f$, $V_{arg}^f$, and $V_{out}^f$ that contain parameters (of the function), return values, arguments (passed to invoked functions), and output values in a function $f$, respectively, which can be easily derived from CFG. Two dashed

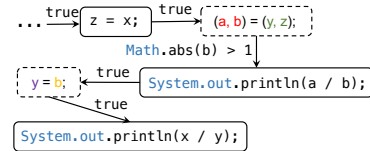

Figure 3: An example of CFG

boxes in Figure 3 show the assignments from arguments to parameters and from the return value to the output value. Based on CFG, we can examine how a program value propagates via *dataflow facts*.

**Definition 2.** (**Dataflow Fact**) There is a dataflow fact from the variable $a$ at line $m$ to the variable $b$ at $b$ at line $n$, denoted by $a@\ell_m \hookrightarrow b@\ell_n$, if the value of $a$ can affect the value of $b$.

In Figure 1, the variable $z$ is assigned with the value of the variable $x$ at line 12 and used as the second argument at line 13. Hence, we have $x@\ell_9 \hookrightarrow x@\ell_{12}$, $z@\ell_{12} \hookrightarrow z@\ell_{13}$, and $x@\ell_9 \hookrightarrow z@\ell_{13}$. Dataflow facts are crucial for many downstream tasks, such as bug detection [4] and program slicing [1]. Specifically, sensitive information leakage, which may cause XSS bugs [21], can be detected by identifying dataflow facts from sensitive data to leaked data. For other bug types, additional restrictions may be imposed on dataflow facts. In the DBZ detection, for example, we constrain that a dataflow fact connects two equal values.

The execution of a statement $s$ can be guarded by a condition. A precise analysis should be *path sensitive*, i.e., validating the feasibility of a fact-inducing path according to the *path condition*.

**Definition 3.** (**Path Condition**) The path condition of a program path $p := s_{i_1} s_{i_2} \cdots s_{i_n}$ is $\psi(p) = \bigwedge_{1 \leq j \leq n-1} E_\ell(s_{i_j}, s_{i_{j+1}})$, where $s_{i_j}$ is the statement at line $i_j$ and can be executed just before $s_{i_{j+1}}$.

We have the dataflow fact $x@\ell_9 \hookrightarrow b@\ell_4$ in Figure 1. However, the statement at line 4 is guarded by `Math.abs(b) > 0`. It is evaluated to be false as the parameter $b$ is passed with 0. Hence, the dataflow fact $x@\ell_9 \hookrightarrow b@\ell_4$ cannot occur in any concrete execution. A path-insensitive analysis would introduce a false positive at line 4 in the DBZ detection.

**Our Problem.** In real-world downstream applications, dataflow analysis focuses on the dataflow facts between specific kinds of variables referred to as *sources* and *sinks*. In the DBZ detection [22], for example, we need to identify the variables that could potentially yield zero values as sources and set the divisors as sinks. Lastly, we formulate the **dataflow analysis problem** as follows.

> Given the source code of a program $P$ and its CFG $G$, identify the dataflow facts from user-specified sources $v_{src} \in V_{src}$ and sinks $v_{sink} \in V_{sink}$ in a path-sensitive manner.

As demonstrated in Section 1, existing dataflow analysis techniques [23, 24, 4] heavily rely on compilation and pose the difficulty of customization, which hinders their applicability and usability [6]. Although several machine learning-based bug detection techniques [25, 26] enable compilation-free analysis, they cannot answer a general dataflow analysis question formulated above and struggle to support the customization for different bug types without a large training dataset. To fill the research

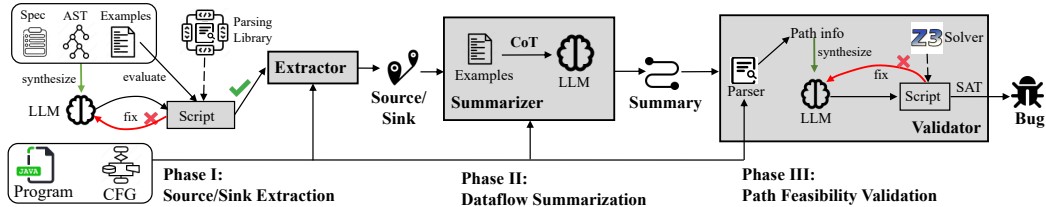

Figure 4: The workflow of LLMDFA consists of three phases

gap, we aim to propose a new paradigm of dataflow analysis in this work. Specifically, we realize the exceptional performance of LLMs in program comprehension [27, 28], highlighting the potential for identifying dataflow facts. Besides, LLMs demonstrate a strong capability of understanding natural language [29, 30], and thus, can effectively comprehend the developers' intents based on natural language descriptions that specify sources and sinks. Moreover, LLMs posses remarkable program synthesis capabilities that facilitate the synthesis of tools invoking external experts to tackle domain-specific problems. Inspired by these observations, we attempt to instantiate dataflow analysis without laborious compilation steps and intricate customization procedures by harnessing the power of LLMs and domain-specific experts.

## 3  Method

We propose LLMDFA, a compilation-free and customizable dataflow analysis, which takes a program and its CFG as input. To resolve hallucinations, we split the analysis into three phases in Figure 4.

- **Source/Sink Extraction:** For a dataflow analysis application, such as DBZ detection, LLMDFA first extracts sources and sinks, which are the start and end points of dataflow facts of our interests.
- **Dataflow Summarization:** Based on the extracted sources ($V_{src}^f$) and sinks ($V_{sink}^f$) of a given function $f$, LLMDFA identifies the dataflow facts from $v \in V_{src}^f \cup V_{par}^f \cup V_{out}^f$ to $v' \in V_{sink}^f \cup V_{arg}^f \cup V_{ret}^f$ as summaries, which form inter-procedural dataflow paths from sources to sinks.
- **Path feasibility Validation:** For each dataflow path from sources to sinks, LLMDFA collects its path condition and validates path feasibility, eventually reporting bugs induced by feasible paths.

The rest of this section demonstrates the detailed technical designs in the three phases.

### 3.1  Phase I: Source/Sink Extraction

Extracting sources and sinks is non-trivial with LLMs. First, querying LLMs whether each line contains sources or sinks is expensive. Second, the hallucinations of LLMs may induce incorrect sources and sinks. To tackle these issues, we utilize LLMs to synthesize script programs using parsing libraries as sources/sink extractors. By traversing the abstract syntax tree (AST) of a given program, script programs can identify sources/sinks at a low cost, yielding deterministic and explainable results.

As shown in the left part of Figure 4, the synthesis requires a specification $\mathcal{S}$ depicting sources/sinks, example programs $\mathcal{E}_{spec}$ with sources/sinks, and their ASTs $\mathcal{T}$. Given a phase description $\mathcal{D}_1$, an extractor $\alpha_E := \alpha_E^{(t)}$ is generated by the conditional probability $p_\theta$ after a specific number of fixes:

$$\alpha_E^{(0)} \sim p_\theta(\cdot \mid \mathcal{D}_1, \mathcal{S}, \mathcal{E}_{spec}, \mathcal{T}) \tag{1}$$

$$\alpha_E^{(i)} \sim p_\theta(\cdot \mid \mathcal{D}_1, \mathcal{S}, \mathcal{E}_{spec}, \mathcal{T}, \mathcal{O}^{(i-1)}), \ 1 \le i \le t \tag{2}$$

$$\Phi(\alpha_E^{(i)}, \mathcal{E}_{spec}) = \chi_{\{t\}}(i), \ 0 \le i \le t \tag{3}$$

Here $\chi_{\{t\}}(i)$ checks whether $i$ is equal to $t$. $\Phi(\alpha_E^{(i)}, \mathcal{E}_{spec}) = 1$ if and only if the script synthesized in the $i$-th round identifies sources and sinks in $\mathcal{E}_{spec}$ with no false positives or negatives. As formulated by Equations (1)$\sim$(3), LLMDFA iteratively fixes a script, utilizing the execution result (denoted by $\mathcal{O}^{(i-1)}$) of the script synthesized in the previous round, until the newly synthesized one correctly identifies sources and sinks in the example programs. Notably, our extractor synthesis is a one-time effort. The synthesized extractors can be reused when analyzing different functions. As an example, Figure 12 in Appendix A.2.3 shows an example program with sources/sinks and the synthesized sink extractor for DBZ detection. The code highlighted in grey is generated by LLMs, while the rest is the skeleton provided manually. We also list our prompt template in Figure 9 of Appendix A.2.2.

## 3.2 Phase II: Dataflow Summarization

We realize that an inter-procedural dataflow fact is the concatenation of multiple intra-procedural dataflow facts from $v \in V_{src}^f \cup V_{par}^f \cup V_{out}^f$ to $v' \in V_{sink}^f \cup V_{arg}^f \cup V_{ret}^f$ in single functions $f$. By identifying intra-procedural dataflow facts as function summaries, we can significantly reduce the prompt length. Besides,

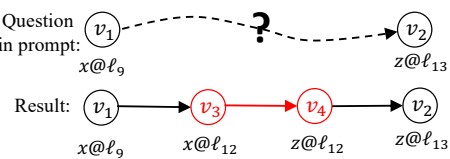

Figure 5: A summary discovered via CoT

a summary can be induced by one or more operations with specific patterns, such as direct uses and assignments. For example, the summary $x@\ell_9 \hookrightarrow z@\ell_{13}$ in Figure 5 is introduced by the assignment at line 12 and the direct use of the variable $z$ at line 13. Offering few-shot examples with detailed explanations would expose typical patterns of dataflow facts that form function summaries, which can align LLMs with program semantics, prompting their ability in dataflow summarization.

Based on the insight, we introduce a few-shot CoT prompting to facilitate the dataflow summarization, which corresponds to the middle part of Figure 4. Given a phase description $\mathcal{D}_2$ and a list of examples with explanations $\mathcal{E}_{flow}$, the response can be generated by the conditional probability:

$$r \sim p_\theta(\cdot \mid \mathcal{D}_2, \ \mathcal{E}_{flow}, \ v, \ v', \ P) \tag{4}$$

where $v \in V_{src}^f \cup V_{par}^f \cup V_{out}^f$, $v' \in V_{sink}^f \cup V_{arg}^f \cup V_{ret}^f$, and $P$ is the program under the analysis. Based on the response $r$, we can determine the existence of the dataflow fact between $v$ and $v'$. Concretely, we construct the prompt according to the template shown in Figure 10 of Appendix A.2.2. Particularly, the examples cover the typical patterns of dataflow facts, and meanwhile, the explanations depict reasoning process in detail. Both designs are pretty crucial for the few-shot CoT prompting. Lastly, we ask LLMs to reason step by step and offer an explanation along with the Yes/No answer.

Consider $x@\ell_9$ and $z@\ell_{13}$ in Figure 1. As shown in Figure 5, LLMDFA discovers intermediate program values, i.e., $x@\ell_{12}$ and $z@\ell_{12}$, and eventually obtains the dataflow fact $x@\ell_9 \rightarrow z@\ell_{13}$step by step. Hence, the strategy of few-shot CoT prompting helps LLMs to better align with program semantics, significantly reducing hallucination in reasoning dataflow facts within single functions.

## 3.3 Phase III: Path Feasibility Validation

Validating path feasibility is a complicated reasoning task that involves determining the satisfiability of a path condition. Although LLMs cannot achieve adequate performance in complex reasoning tasks [14], several off-the-shelf domain-specific experts, such as SMT solvers, can be utilized by LLMs. For example, the Python binding of Z3 solver [15] enables us to solve constraints in Python code. Hence, we propose to synthesize a Python script program that encodes and solves path conditions according to path information. Decoupling constraint solving from path condition collection can substantially mitigate the hallucination in path feasibility validation.

The right part of Figure 4 shows the workflow of the path feasibility validation. Based on the dataflow facts stitched from summaries, LLMDFA first leverages a parser to extract the path information, such as the branches exercised by the program path and the branch conditions. Notably, the parser receives the program lines appearing in the summaries as inputs and does not need to be reimplemented for different forms of sources and sinks. Based on the derived path info $\mathcal{I}$ and the phase description $\mathcal{D}_3$, a script program $\alpha_V := \alpha_V^{(t)}$ is eventually generated after a specific number of fixes as follows:

$$\alpha_V^{(0)} \sim p_\theta(\cdot \mid \mathcal{D}_3, \ \mathcal{I}) \tag{5}$$

$$\alpha_V^{(i)} \sim p_\theta(\cdot \mid \mathcal{D}_3, \ \mathcal{I}, \ err^{(i-1)}), \ 1 \le i \le t \tag{6}$$

$$err^{(t)} = \epsilon, \ err^{(i)} \ne \epsilon, \ 0 \le i \le (t-1) \tag{7}$$

Particularly, we utilize the error message of executing the script synthesized in a previous round ($(i-1)$-th round), denoted by $err^{(i-1)}$, and feed it to LLMs to conduct the fixing in the $i$-th round. Concretely, we design the prompt template shown by Figure 11 in Appendix A.2.2. It should be noted that the synthesis process has to be repeated for each source-sink pair, which is different from the one-time effort paid in the extractor synthesis. Consider $x@\ell_9 \hookrightarrow b@\ell_4$ in Figure 1. We offer the branch condition `Math.abs(b) > 1` and other path information in a prompt and obtain a script in Figure 6. To ease the synthesis, we offer

```python
from z3 import *
s = Solver()
b = Int('b')
s.add(b == 0)
s.add(Abs(b) > 1)
print(s.check())
```

Figure 6: A script invoking Z3 solver

Table 1: The performance of LLMDFA in the overall detection and the three phases when using different LLMs. **P**, **R**, and **F1** indicate the precision, recall, and F1 score, respectively.

| Bug | Phase | gpt-3.5 | | | gpt-4 | | | gemini-1.0 | | | claude-3 | | |
|---|---|---|---|---|---|---|---|---|---|---|---|---|---|
| | | **P** (%) | **R** (%) | **F1** | **P** (%) | **R** (%) | **F1** | **P** (%) | **R** (%) | **F1** | **P** (%) | **R** (%) | **F1** |
| DBZ | Extract | 100.00 | 100.00 | 1.00 | 100.00 | 100.00 | 1.00 | 100.00 | 100.00 | 1.00 | 100.00 | 100.00 | 1.00 |
| | Summarize | 90.95 | 97.57 | 0.94 | 95.32 | 98.43 | 0.97 | 83.57 | 82.47 | 0.83 | 89.26 | 92.38 | 0.91 |
| | Validate | 81.58 | 99.20 | 0.90 | 89.76 | 100.00 | 0.95 | 79.83 | 93.73 | 0.86 | 85.74 | 94.52 | 0.90 |
| | **Detection** | **73.75** | **92.16** | **0.82** | **81.38** | **95.75** | **0.87** | **66.57** | **74.21** | **0.70** | **76.91** | **82.67** | **0.80** |
| XSS | Extract | 100.00 | 100.00 | 1.00 | 100.00 | 100.00 | 1.00 | 100.00 | 100.00 | 1.00 | 100.00 | 100.00 | 1.00 |
| | Summarize | 86.52 | 96.25 | 0.91 | 97.84 | 99.76 | 0.99 | 88.79 | 97.31 | 0.93 | 94.17 | 97.83 | 0.96 |
| | Validate | 100.00 | 100.00 | 1.00 | 100.00 | 98.91 | 0.99 | 100.00 | 99.07 | 1.00 | 100.00 | 95.29 | 0.98 |
| | **Detection** | **100.00** | **92.31** | **0.96** | **100.0** | **98.64** | **0.99** | **100.00** | **94.60** | **0.97** | **100.0** | **86.49** | **0.93** |
| OSCI | Extract | 100.00 | 100.00 | 1.00 | 100.00 | 100.00 | 1.00 | 100.00 | 100.00 | 1.00 | 100.00 | 100.00 | 1.00 |
| | Summarize | 89.57 | 85.76 | 0.88 | 94.58 | 93.12 | 0.94 | 87.21 | 96.54 | 0.92 | 98.26 | 97.87 | 0.98 |
| | Validate | 100.00 | 97.14 | 0.99 | 100.00 | 100.00 | 1.00 | 100.00 | 98.13 | 0.99 | 100.00 | 100.00 | 1.00 |
| | **Detection** | **100.00** | **78.38** | **0.88** | **100.00** | **89.19** | **0.94** | **100.00** | **94.59** | **0.97** | **100.00** | **97.30** | **0.99** |

lines 1, 2, and 6 in a skeleton. We refine the script at most three times. If the script is buggy after three trials, LLMDFA enforces LLMs to determine path feasibility based on the path information.

## 4 Evaluation

We implement LLMDFA as a prototype analyzing Java programs. Utilizing the parsing library `tree-sitter` [31], LLMDFA obtains the parameters, return values, callers/callees, and sources/sinks. We configure LLMDFA with four LLMs across various architectures, namely `gpt-3.5-turbo-0125`, `gpt-4-turbo-preview`, `gemini-1.0-pro`, and `claude-3-opus`. We proceeded to evaluate the performance of LLMDFA through extensive experiments, with the total cost amounting to 1622.02 USD. In the rest of the paper, we use `gpt-3.5`, `gpt-4`, `gemini-1.0`, and `claude-3` for short without ambiguity. To reduce the randomness, we set the temperature to 0 so that LLMDFA performs greedy decoding without any sampling strategy.

### 4.1 Dataset

**Synthetic Benchmark.** Juliet Test Suite [17] is a benchmark widely used to evaluate static analyzers. Considering the high impact and representativeness, we choose divide-by-zero (DBZ), cross-site-scripting (XSS), and OS Command Injection (OSCI) for evaluation. As illustrated in Section 2, the dataflow facts inducing DBZ bugs are more restrictive than the ones inducing XSS bugs, as the values in the former should be equal instead of just being dependent. The OSCI bugs share the same forms of dataflow facts as XSS bugs, while their sources and sinks have different forms. Indicated by comments, there are 1,850 DBZ, 666 XSS, and 444 OSCI bugs in total. To avoid the leakage of ground truth, we remove comments and obfuscate code before the evaluation.

**Real-World Programs.** We choose TaintBench Suite [19], which consists of 39 real-world Android malware applications. Due to the reliance on Gradle in an old version, we cannot compile the applications in our environment. Besides, each application is equipped with specific sources and sinks that are customized to its unique functionality. For instance, if the argument of the function *startService* depends on the return value of the function *getDisplayOriginatingAddress*, it may result in the leakage of user address information. The highly customized sources and sinks require us to tailor dataflow analysis for each application. Therefore, TaintBench serves as ideal subjects for evaluating LLMDFA in a compilation-free and customizable scenario.

### 4.2 Performance of LLMDFA

**Setup and Metrics.** We evaluate LLMDFA upon Juliet Test Suite to measure its precision, recall, and F1 score. Apart from the overall detection, we also measure the performance of each phase to quantify its effectiveness. Specifically, we diff the sources/sinks labeled in the benchmark and the identified ones to measure the performance of source/sink extraction. Besides, we measure the precision and recall of dataflow summarization by manually examining the value pairs investigated by LLMDFA. Lastly, we manually examine generated path conditions and compute the precision and recall of identifying feasible paths. Due to the lack of explicit ground truth in the benchmark, we would

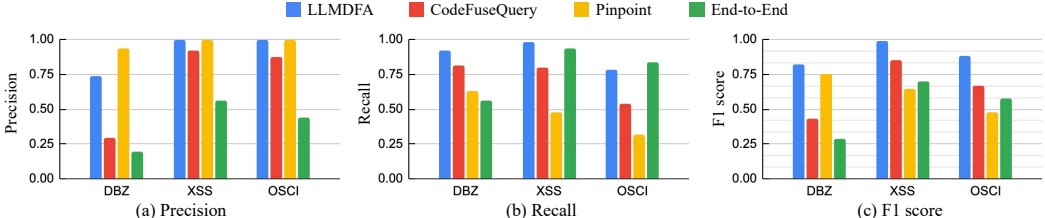

Figure 7: The comparison of LLMDFA, CodeFuseQuery, Pinpoint, and LLM-based end-to-end analysis. LLMDFA and LLM-based end-to-end analysis are powered with `gpt-3.5`

have to make laborious efforts to examine thousands of program paths. To simplify examination, we choose 37 programs for each bug type to measure the performance of the last two phases, as other programs only differ from the selected ones in terms of sources and sinks.

**Result.** As shown in Table 1, LLMDFA achieves high performance in each phase and the overall detection when it is powered with different LLMs. Equipped with `gpt-3.5`, for example, it achieves the precision of 73.75%/100%/100%, the recall of 92.16%/92.31%/78.38%, and the F1 score of 0.82/0.96/0.88, in the DBZ/XSS/OSCI detection. Besides, LLMDFA synthesizes all the source/sink extractors successfully for the three bug types. In the DBZ/XSS/OSCI detection, the dataflow summarization achieves 90.95%/86.52%/89.57 precision and 97.57%/96.25%/85.76% recall, and meanwhile, the path feasibility validation achieves 81.58%/100.00%/100.00% precision and 99.20%/100.00%/97.14% recall. When utilizing other LLMs, LLMDFA achieves the precision and recall comparable to those obtained using `gpt-3.5`. While the precision of the DBZ detection powered by `gemini-1.0` is slightly lower at 66.57% than other LLMs, the performance remains satisfactory, exhibiting superiority over the baselines in Section 4.3. LLMDFA successfully synthesizes the extractors and script programs encoding path conditions with only a few iterations, which is demonstrated in Appendix A.3.1 and A.3.2. Lastly, it is found that the average financial costs of the DBZ, XSS, and OSCI detection are 0.14 USD, 0.05 USD, and 0.04 USD, respectively. Such a cost is in line with works of a similar nature, such as [32], which takes 0.42 USD to repair a bug. Notably, the extractor synthesis is one-time for a given bug type. Hence, the financial cost of the detection in practice is even lower. Overall, the statistics show the generality, effectiveness, and efficiency of LLMDFA in detecting dataflow-related bugs.

## 4.3 Comparison with Baselines

**Classical Dataflow Analysis.** We choose two industrial static analyzers, namely CodeFuseQuery [18] and Pinpoint [4], for comparison. Specifically, CodeFuseQuery does not depend on any compilation process and derives dataflow facts from the ASTs of programs, while Pinpoint requires the compilation and takes as input the LLVM IR generated by a compiler. As shown by Figure 7, CodeFuseQuery achieves 29.41%/92.26%/87.46% precision, 81.08%/79.67%/54.05% recall, and 0.43/0.86/0.67 F1 score in detecting the DBZ/XSS/OSCI bugs. The low precision of the DBZ detection is attributed to its path-insensitive analysis. The low recall of CodeFuseQuery is caused by the lack of the support of analyzing complex program constructs. For example, the inability to analyze global variables causes missing dataflow facts, which causes more false negatives.

Besides, Pinpoint obtains 93.78%/100.00%/100.00% precision, 63.19%/47.49%/31.35% recall, and 0.76/0.64/0.48 F1 score in the detection of DBZ/XSS/OSCI bugs. Due to the lack of comprehensive modeling and customized support for source and sink, Pinpoint misses a large number of buggy dataflow paths, which eventually result in low recall. Although it achieves a high precision of 93.78% in DBZ detection, LLMDFA demonstrates superior capabilities in detecting DBZ bugs, delivering not only satisfactory precision but also significantly higher recall and F1 scores.

**LLM-based End-to-End Analysis.** We adopt few-shot CoT prompting to detect the DBZ, XSS, and OSCI bugs. Specifically, we construct few-shot examples to cover all the forms of sources and sinks, and meanwhile, explain the origin of a bug step by step. Figure 7 shows the performance comparison when using `gpt-3.5`. LLMDFA exhibits superiority over LLM-based end-to-end analysis in detecting three kinds of bugs. Although the recall of LLMDFA is slightly lower than the baseline in the OSCI detection, the precision of the former is much higher than the latter. We obtain the same findings

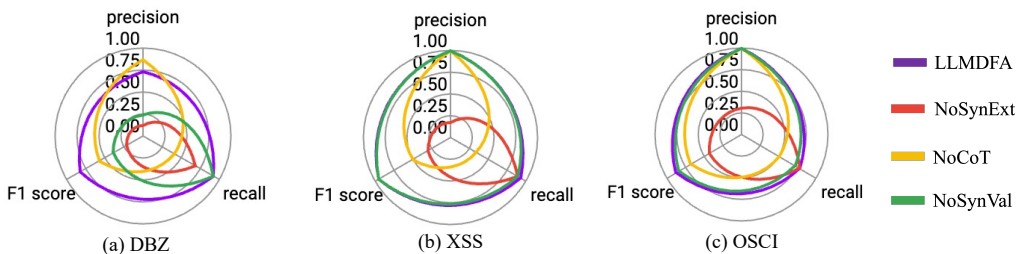

Figure 8: The comparison of LLMDFA and ablations using `gpt-3.5`

from the results of the analysis powered by the other three LLMs, which are shown by Figure 14 in Appendix A.3.3. In Appendix A.4.1, we offer typical cases where the LLM-based end-to-end analysis identifies wrong dataflow facts due to hallucinations.

## 4.4 Ablation Studies

**Setup and Metrics.** We introduce three ablations, namely NoSynExt, NoCoT, and NoSynVal and measure their performance in detecting the DBZ, XSS, and OSCI bugs. Specifically, NoSynExt directly leverages LLMs to extract sources and sinks. NoCoT provides the descriptions of dataflow facts in prompts and summarizes dataflow facts without few-shot CoT prompting.NoSynVal validates path feasibility with LLMs directly without synthesizing programs invoking SMT solvers.

**Result.** Figure 8 shows the comparison results of the ablations using `gpt-3.5`. First, LLMDFA has an overwhelming superiority over NoSynExt and NoCoT. Although NoCoT achieves 86.8% precision while LLMDFA obtains 73.7% precision in the DBZ detection, LLMDFA has much larger recall and F1 score than NoCoT. The key reason is that NoCoT is unable to identify complex dataflow facts, which causes the low recall of NoCoT. Second, NoSynExt introduces false positives in many cases because of the low precision of source/sink extraction in both the DBZ, XSS, and OSCI detection. Third, it should be noted that LLMDFA does not show significant superiority over NoSynVal in the XSS and OSCI detection because the corresponding benchmark programs do not contain any XSS or OSCI bug-inducing infeasible paths. Also, LLMDFA may encode the path condition incorrectly and accept the infeasible path, eventually causing false positives. As shown by Figure 15 in Appendix A.3.4, We can obtain similar findings when using other three LLMs. Although the precisions vary among different LLMs and bug types, LLMDFA is always superior to the ablations, demonstrating its effectiveness in mitigating the hallucinations in LLM-powered dataflow analysis. We also offer several examples of mitigated hallucinations in Appendix A.4.1.

## 4.5 Evaluation upon Real-world Programs

Table 2 shows the basic statistics of TantBench Suite [19]. Particularly, there are 53 different forms of source-sink pairs in total and an application contain 7.9 different forms of source-sink pairs on average. The diversity of source-sink pairs necessitates the customization of dataflow analysis. Considering the resource cost of invoking LLMs and the manual cost of comparing detection results with the ground truth, we run LLMDFA with `gpt-3.5` for randomly selected source/sink pairs, which constitute 80 out of the 203 dataflow paths in the ground truth. As shown in Table 3, LLMDFA achieves the precision of 75.38% and the recall of 61.25%. The false positives and negatives are mainly caused by the hallucinations in the dataflow summarization. Listing 8 in Appendix A.4.3 shows a false negative example when LLMDFA fails to detect the dataflow path from the return value of the function *query* to the argument of the function *write*. Specifically, LLMDFA fails to capture the summary of *processResults*, which contains complex program structures, such as a try-catch block and a while statement, ultimately resulting in the failure to report the intended dataflow path. We also present a false positive example by Listing 8 Appendix A.4.3.

We also compare LLMDFA with the end-to-end analysis and CodeFuseQuery. Specifically, we adopt a few-shot CoT prompting strategy to conduct the end-to-end analysis and customize CodeFuseQuery by designing specific queries to capture dataflow paths accordingly. As shown by Table 3, the end-to-end analysis only achieves a precision of 43.48% and a recall of 25.00%. The precision and recall are primarily affected by the incorrect identification of sources and sinks. When dealing

Table 2: The statistics of TaintBench, including the total number of program lines, functions, and source-sink pairs, along with the maximal and average number per application.

| Metric | Total | Max | Avg |
|---|---|---|---|
| Program Line | 1,376,214 | 260,797 | 35,287.03 |
| Function | 155,364 | 23,414 | 3,983.69 |
| Source-sink Pair | 53 | 17 | 7.9 |

Table 3: The performance results upon Taint-Bench. A1~A3 are LLMDFA with `gpt-3.5`, end-to-end analysis with `gpt-3.5`, and Code-FuseQuery, respectively.

| Analysis | Precision | Recall | F1 Score |
|---|---|---|---|
| A1 | **75.38%** | **61.25%** | **0.67** |
| A2 | 43.48% | 25.00% | 0.32 |
| A3 | 72.92% | 43.75% | 0.55 |

Table 4: The performance of LLMDFA upon Juliet Test Suite of C/C++ version

| | Precision | Recacc | F1 Score |
|---|---|---|---|
| DBZ | 85.71% | 83.94% | 84.77% |
| APT | 100.00% | 86.83% | 92.92% |
| OSCI | 97.36% | 73.68% | 83.98% |

Table 5: The performance of LLMDFA upon the real-world benchmark SecBench.js

| Bug Type | Precision | Recacc | F1 Score |
|---|---|---|---|
| Command Injection | 94.44% | 67.33% | 0.79 |
| Tainted Path | 50.00% | 100.00% | 0.67 |
| Code Injection | 90.91% | 83.33% | 0.87 |
| **Overall** | **92.52%** | **71.74%** | **0.81** |

with large programs, a lengthy prompt can amplify the occurrence of false positives/negatives in identifying sources/sinks and discovering dataflow facts between them. While CodeFuseQuery achieves comparable precision to LLMDFA reaching 72.92%, its recall is significantly lower at only 43.75% compared to LLMDFA. Its low recall is mainly attributed to its limited ability to analyze complex program constructs such as global variables and arrays. This observation is consistent with the evaluation results obtained upon Juliet Test Suite. We do not evaluate the compilation-dependent analyzer Pinpoint because the subjects in TaintBench cannot be compiled successfully due to an outdated version of Gradle. Overall, the above statistics provide sufficient evidence of the potential of LLMDFA in customizing dataflow analysis for real-world projects without compilation.

## 4.6 Multi-linguistic Support

Although the multi-linguistic support is not our main contribution, it is easy to extend LLMDFA to support other languages due to its compilation-free design. Based on dataflow analysis theory, we only need to construct the control flow graph with corresponding tree-sitter packages and then reuse the current implementation. In our evaluation, we further migrate Java analysis to support C/C++/JavaScript and evaluate it with `gpt-3.5` upon the Juliet Test Suite for C/C++ and a real-world JavaScript benchmark SecBench.js [33]. The Juliet Test Suite for C/C++ does not contain XSS bugs. Hence, we chose another bug type, namely Absolute Path Traversal (APT), which is a typical security vulnerability. As shown in Table 4, the performance of LLMDFA on C/C++ is equally good as on Java. SecBench.js includes 138 vulnerabilities in JavaScript packages, covering command injection, taint path, and code injection vulnerabilities, which have been assigned with CVE IDs due to significant security impact. Table 5 shows that our compilation-free analysis upon SecBench.js achieves 92.54% precision and 71.74% recall, which are comparable with the compilation-dependent approach in [33]. Notably, the migration only requires the modification of no more than 100 lines of code and mostly relates to the changes to AST node types and parser initialization.

## 4.7 Limitations and Future Works

First, the prompts can be lengthy in the few-shot CoT prompting, which can result in significant token cost. Frequent prompting can also increase time overhead. Hence, LLMDFA is better suited for analyzing specific program modules than the entire program. To make whole program analysis practical, we may need to accelerate the inference or parallelize LLMDFA. Second, dataflow summarization can be imprecise in the presence of large functions or sophisticated pointer operations, further leading to incorrect results in identifying dataflow facts across functions. A potential improvement is to fine-tune existing LLMs with the dataflow facts produced by classical dataflow analyzers. Third, LLMDFA may not accurately encode path conditions and potentially compromise the soundness. Appendix A.4.2 presents several cases where LLMDFA encodes path conditions wrongly. It is possible to investigate several patterns of path conditions and synthesize script programs to over-approximate them, which could discard infeasible paths while retaining high recall simultaneously.

# 5  Related Work

**Dataflow Analysis.** Current dataflow analysis predominantly relies on IR code generated by semantic analysis during the compilation, such as LLVM IR [5] and Soot IR [34]. Typically, SVF [24] and Klee [35] analyze C/C++ programs based on LLVM IR code. Industrial analyzers like Infer [36] and Semmle [23] also require successful builds to obtain necessary IR code for analysis. Consequently, the reliance on compilation infrastructures restricts the applicability when target programs cannot be compiled. Besides, existing techniques abstract semantics with formal structures, such as graphs [1] and logical formulas [37], to compute dataflow facts of interests. However, semantic abstraction differs greatly depending on analysis demands, such as the choices of sources/sinks and the precision setting of the analysis. Hence, the customization of dataflow analysis requires laborious manual effort and expert knowledge, which hinders its widespread adoption in real-world scenarios [6].

**Machine Learning-based Program Analysis.** The first line of studies derives program properties, such as library specifications [38, 39] and program invariants [40, 41], to augment classical analyzers. For instance, USpec utilizes large codebases to predict aliasing relation [38]. SuSi employs classification models to infer the sources and sinks of sensitive information [39]. While these techniques offer analyzers insightful guidance, they do not have any correctness guarantees due to their inherent limitations. The second line of works targets data-driven bug detection with training models [25, 42, 43]. Typically, [25] jointly trains an embedding model and a classification model upon a large training dateset. Unlike LLMDFA, it cannot answer general dataflow queries upon two specific program values. Similarly, Hoppity detects JavaScript bugs with the model trained upon a large volume of buggy code [43]. The reliance to training data make these techniques difficult to customize for specific analysis demands in the presence of a sufficient amount of training data.

**LLMs for Program Analysis.** The emergence of LLMs has created exciting opportunities for various analysis tasks, including code completion [44], repair [9, 45], and comprehension [46]. Considerable research targets the reasoning abilities of LLMs through techniques such as CoT [16], ToT [47], and accumulative reasoning [14]. However, only a few studies focus on domain-specific reasoning for programs. Typically, several studies employ the CoT prompting to infer program invariants [10, 48] and rank potential invariants [49]. Also, LLift retrieves function specifications through prompting [50] to assist classical bug detectors. As far as we know, no previous studies have solely relied on LLMs for program analysis. Our work formulates chain-like structures in dataflow facts and the demonstrates the possibility of synthesizing new tools to avoid hallucinations in program analysis. Our insight into mitigating hallucinations can be generalized to other software engineering problems, such as program repair [45] and program synthesis [51, 52].

# 6  Conclusion

This paper presents LLMDFA, a LLM-powered compilation-free and customizable dataflow analysis. To mitigate the hallucinations, it decomposes the whole analysis into three manageable sub-problems and solves them with a series of strategies, including tool synthesis, few-shot CoT prompting, and formal method-based validation. Our evaluation shows the remarkable performance of LLMDFA in analyzing both synthetic and real-world programs. Our work demonstrates a promising paradigm for reasoning code semantics, with the potential for generalization to other code-related tasks.

## Acknowledgement

We are grateful to the Center for AI Safety for providing computational resources. This work was funded in part by the National Science Foundation (NSF) Awards SHF-1901242, SHF-1910300, Proto-OKN 2333736, IIS-2416835, DARPA VSPELLS - HR001120S0058, IARPA TrojAI W911NF-19-S0012, ONR N000141712045, N000141410468 and N000141712947. Any opinions, findings and conclusions or recommendations expressed in this material are those of the authors and do not necessarily reflect the views of the sponsors.

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

# A  Appendix

## A.1  Broader Impact

This paper presents work whose goal is to advance the field of machine learning, targeting a complicated code-reasoning task, namely dataflow analysis. We have demonstrated the limitations of our work above. We do not expect our work to have a negative broader impact, though leveraging LLMs for code-reasoning tasks may come with certain risks, e.g., the leakage of source code in private organizations and potential high token costs. Meanwhile, it is worth more discussions to highlight that our work has the potential to dramatically change the field of software engineering with the power of LLMs. Specifically, LLM-powered dataflow analysis not only enables the analysis of incomplete programs with little customization but addresses other challenges in classical dataflow analysis.

First, classical dataflow analyzers mainly depend on specific versions of IRs. As a compilation infrastructure evolves, the version of the IR code can change, requiring the implementation to migrate to support the analysis of new IRs. For example, the Clang compiler has undergone ten major version updates in the past decade [53], resulting in differences between IRs generated by different compiler versions. The IR differences necessitate tremendous manual efforts to migrate the dataflow analysis implementation for each version update in the long term. However, an LLM-powered dataflow analysis directly operates on the source code and supports different language standards.

Second, classical dataflow analysis lies in the various abstractions and precision settings [54], especially pointer analysis, a fundamental pre-analysis in the classical dataflow analysis workflow. Specifically, the developers of dataflow analyzers have to consider different precision settings, such as Anderson-style pointer analysis [55] and Steensgaard's pointer analysis [56], and implement the analysis algorithms accordingly. This process demands significant implementation effort. In contrast, LLMs, being aligned with program semantics, serve as interpreters of program semantics and eliminate the need to propose abstractions and implement analysis under specific precision settings. Instead, we can interact with LLMs by prompting them to query the program facts of interest very conveniently.

## A.2  Prompt of LLMDFA and Baseline

This section presents more details of prompt design in LLMDFA and the end-to-end analysis, including the basic principle, prompt templates, and the forms of the examples used in few-shot prompting.

### A.2.1  Prompt Design Principle of LLMDFA

To facilitate the synthesis of src/sink extractors, we enumerate the various types of sources and sinks derived from the definition of dataflow-related bugs. Specifically, we follow the comments in Juliet Test Suite and manually specify different forms of sources and sinks. To support dataflow summarization, we present illustrative examples showcasing different dataflow patterns, including the dataflow facts resulting from direct uses, assignments, and load/store operations upon pointers. Regarding the script synthesis for path feasibility validation, we do not offer examples and instead provide a basic skeleton of a Python script program. We then prompt LLMs to complete the script based on the provided path information.

### A.2.2  Prompt Templates of LLMDFA

The detailed prompt templates used in the DBZ detection are shown in Figure 9, Figure 10, and Figure 11. For other bug types, the prompt templates are similar. Notably, all the examples in the prompts consist of no than six lines of code. When writing the examples and explanations, we maintain a consistent program structure and sentence format, which can be easily achieved via simple copy-and-paste operations and minor modifications upon specific statements or expressions. This process requires no specialized expertise and entails minimal manual effort.

### A.2.3  Example Program and Sink Extractor for DBZ Detection

Figure 12 shows the example program and the synthesized sink extractor in the DBZ detection.

| |
|---|
| **Role:** You are a good programmer and familiar with AST of programs. 
 **Description**: Please write the Python script traversing AST and identify sources/sinks for data flow analysis. |
| **Source/Sink Info:** There are several forms of sources/sinks: [Spec]. Also, we offer several example programs containing sources/sinks and their corresponding ASTs: [Example Programs + ASTs] |
| **Synthesis Task:** Please write a Python script to extract the sources/sinks on AST. You may refer to the AST structure of the example programs, and a skeleton AST traverser program. [skeleton] |
| **Fixing Task:** Here is the synthesized result of last round: [script]. When executing the script, we encounter the following error: [error message]. Here are missed sources/sinks are missed: [missed ones]. Here are the variables misidentified as sources/sinks: [incorrect ones]. Please fix it and return a runnable script. |

Figure 9: The prompt template of the source/sink extractor synthesis in the DBZ detection

| |
|---|
| **Role:** You are a good Java programmer. You are good at understanding the semantics of Java programs. 
 **Description:** Determine whether two variables at two lines have the same value. |
| Here are several **rules**: 
 (1) If they are the same variable and not overwritten between two lines, the answer should be yes. 
 (2) If the variable a is assigned with the value of the variable b, then answer should be yes. [Other rules] |
| Here are several **examples**: 
 Example 1: User: [Program] [Question] 
 System: [Answer: Yes] [Explanation: y is assigned with x at line 2 and not over-written between lines 2 and 3. Hence, the value of y at line 3 is the same as x defined at line 1. The answer is Yes.] 
 [Other examples] |
| **Question:** Now I give you a function: [FUNCTION] 
 Please answer: Does [VAR1] used at line [L1] have the same value as [VAR2] defined at line [L2] ? 
 Please think it step by step. Return Yes/No with the explanation. |

Figure 10: The prompt template of the dataflow summarization in the DBZ detection

| |
|---|
| **Role:** You are a good programmer and familiar with Z3 python binding. 
 **Description**: Please write a Python program using Z3 python binding to encode the path condition. |
| **Path Info:** Here is a path: [path]. Note that the value of [variable] is 0. Line [line number] is in the [true/false] branch of the if-statement, of which the condition is [branch condition]. |
| **Synthesis Task:** Please write a Python script to solve the path condition using Z3 python binding. You can refer to the skeleton: [skeleton] |
| **Fixing Task:** Here is the synthesized result of last round: [script]. When executing the synthesized script, we encounter the following error: [error message]. Please fix the bug and return a runnable script. |

Figure 11: The prompt template of the path feasibility validation in the DBZ detection

```
int x1 = 0; // src: x1
double x2 = 0.0; // src: x2
float x3 = 0.0f; // src: x3

var s = "0";
int x4 = parseInt(s); //src: x4

int z = 1;
int a = z / x; // sink: x
int b = z % y; // sink: y
int c = x + y;
```
(a)

```
def is_interesting(node):
  return (node.type=="binary_expr"
      and (node.op == "%" or
          node.op == "/"))
def traverse(node, sinks):
  if is_interesting(node):
    sinks.append(node.sec_operand)
  return sinks
  for child in node.children:
    sinks = traverse(child, sinks)
sinks = traverse(ast_root, [])
```
(b)

Figure 12: An example program with sources/sinks (a) and a sink extractor (b) for the DBZ detection

### A.2.4 Prompt Design of LLM-based End-to-end Analysis

To facilitate the LLM-based end-to-end analysis, we include concise code snippets that encompass all forms of sources and sinks in a prompt. Each code snippet represents a single pair of source and sink. Additionally, we provide a concise explanation of the step-by-step dataflow process from source

Table 6: Comparison of performance for different models on various bug types and kinds (Source/Sink). **NFP**(%) indicates the proportion of successful synthesized extractors without any fixes. **MNF** and **ANF** are the maximal number and average number of fixes, respectively.

| Bug | Kind | gpt-3.5 | | | gpt-4 | | | gemini-1.0 | | | claude-3 | | |
|-----|------|---------|----|----|-------|----|----|-----------|----|----|---------|----|----|
| | | NFP(%) | MNF | ANF | NFP(%) | MNF | ANF | NFP(%) | MNF | ANF | NFP(%) | MNF | ANF |
| DBZ | Source | 95.00 | 11 | 0.55 | 95.00 | 1 | 0.05 | 95.00 | 1 | 0.05 | 100.00 | 0 | 0 |
| | Sink | 100.00 | 0 | 0 | 100.00 | 0 | 0 | 95.00 | 1 | 0.05 | 100.00 | 0 | 0 |
| XSS | Source | 100.00 | 0 | 0.00 | 15.00 | 4 | 1.85 | 95.00 | 1 | 0.05 | 100.00 | 0 | 0 |
| | Sink | 15.00 | 4 | 2.47 | 90.00 | 1 | 0.05 | 95.00 | 2 | 0.10 | 0.00 | 30 | 7.20 |
| OSCI | Source | 95.00 | 2 | 0.10 | 40.00 | 2 | 0.65 | 100.00 | 0 | 0 | 95.00 | 11 | 0.55 |
| | Sink | 90.00 | 6 | 0.55 | 95.00 | 1 | 0.05 | 100.00 | 0 | 0 | 100.00 | 0 | 0 |

to sink. In what follows, we present an example from the prompt that supports the few-shot CoT prompting in the end-to-end analysis for the DBZ detection.

Listing 1: An example used in the few-shot CoT prompting-based end-to-end analysis

```
User:
Here is the program:
1.  int x = 0;
2.  x = 1 + x;
3.  int y = x;
4.  int z = 10 / y;
Please detect the DBZ bugs in the program step by step.

System:
There is 1 divide-by-zero (DBZ) bug in the program:
Bug 1: The value of x at the line 1 is assigned by the literal 0. It is used to assign zero value to
        the variable y at line 3. The value of y is used as the divisor at the line 4, such that the
        divisor is equal to 0. Hence, there is a DBZ bug at the line 4.
```

We do not show the template of the prompts used for the LLM-based end-to-end analysis as it is quite similar to the one used in the dataflow summarization shown in Figure 10. The only difference is that we demonstrate different forms of sources and sinks along with various dataflow patterns in the prompts to achieve an end-to-end solution.

## A.3 Additional Experimental Results

This section presents additional experimental results. Specifically, we demonstrate detailed results of the three stages of LLMDFA, such as the fixing numbers of source/sink extractor synthesis and the script program synthesis in the path feasibility validation. Then we offer the performance results of LLMDFA, end-to-end analysis, and the ablations equipped with different LLMs. Due to the page limit, we do not provide them in the main body of our paper.

### A.3.1 Fix Numbers of Source/Sink Extractor Synthesis

In our evaluation, we set the temperature to 0 to enforce the LLMs to perform greedy encoding without any sampling strategy. However, randomness still exists due to GPU's inherent non-determinism. Considering the resource cost of invoking LLMs, we repeatedly synthesize source/sink extractors 20 times. As shown in Table 6, most of the source/sink extractors can be synthesized without any fixes. Even if LLMDFA fails to synthesize the extractors for the first time, such as LLMDFA with `claude-3` has to fix the sink extractors for the XSS detection in most of the cases, the average number of the fixes is only 7.20, indicating that LLMDFA can finish the extractor synthesis without many iterations.

### A.3.2 Fixing Numbers of Path Feasibility Validation

We also quantify the fixing numbers of Python script program synthesis in the path feasibility validation. As shown in Figure 13, 75.20%, 96.43%, and 85.25% of scripts are synthesized using `gpt-3.5` without any fixes in the DBZ, XSS, and OSCI detection, respectively, which is shown in Figure 13 (a). Particularly, only 0.61%, 3.57%, and 4.92% of the synthesized Python scripts fail after three rounds of fixing in the DBZ, XSS, and OSCI detection, respectively, eventually falling back to the strategy of directly utilizing LLMs for the path feasibility validation. It is also found that 78.57%, 88.68%, and 76.34% of synthesized scripts in the DBZ, XSS, and OSCI detection correctly encode the path conditions, respectively. Although several path conditions are encoded incorrectly, their satisfiability remains the same as the original ones. One typical example is that the LLM interprets `Math.abs(b) > 1` in Figure 1 as the constraint `And(b > 1, b < -1)`, while the correct encoding should be `Or(b > 1, b < -1)`. However, such wrongly encoded constraints still enable us to refute infeasible paths. We offer more case studies in Appendix A.3.2. When validating path feasibility with other models, including `gpt-4`, `gemini-1.0`, and `claude-3`, LLMDFA can also synthesize Python scripts as solving programs with a small number of fixes, which are shown by Figure 13 (b)~(d).

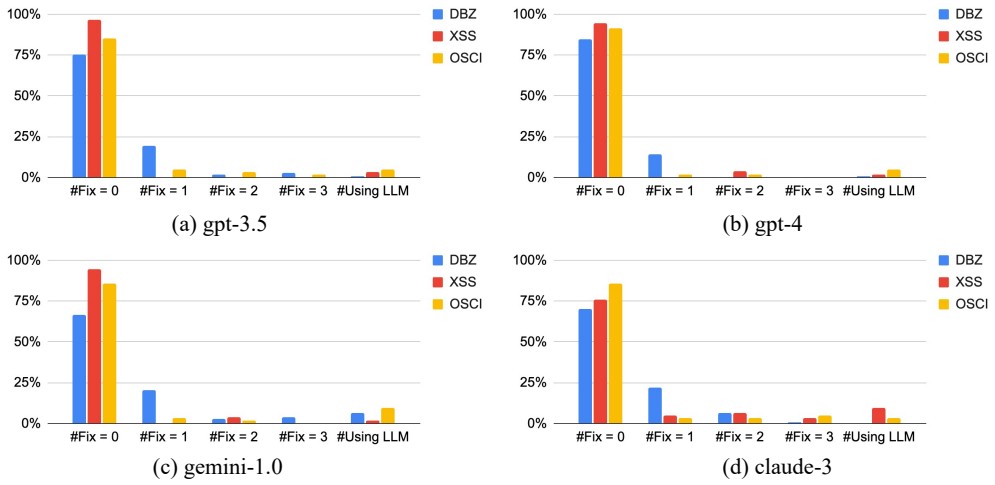

Figure 13: The numbers of fixes in path feasibility validation using different LLMs

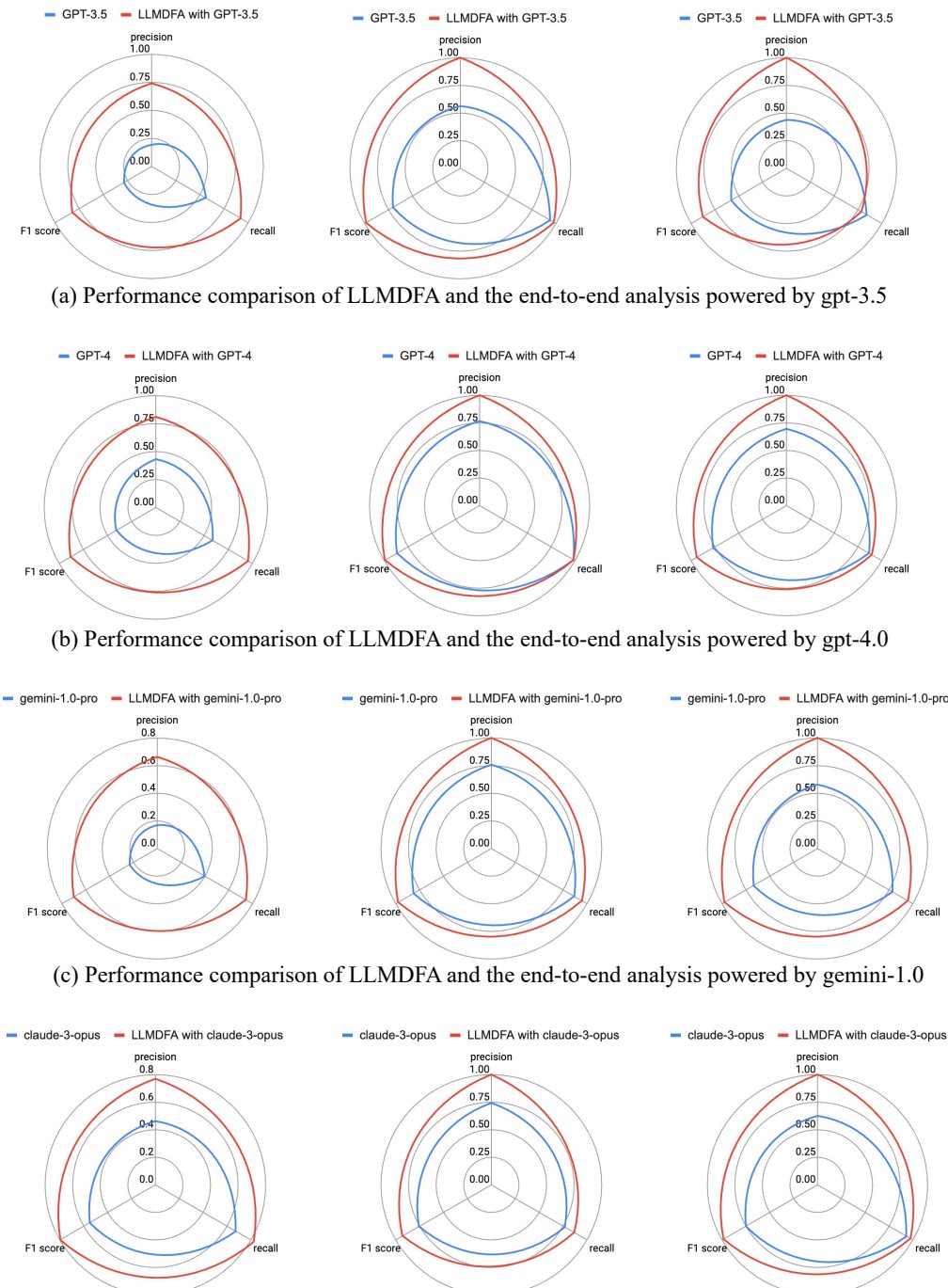

(a) Performance comparison of LLMDFA and the end-to-end analysis powered by gpt-3.5

(b) Performance comparison of LLMDFA and the end-to-end analysis powered by gpt-4.0

(c) Performance comparison of LLMDFA and the end-to-end analysis powered by gemini-1.0

(d) Performance comparison of LLMDFA and the end-to-end analysis powered by claude-3

Figure 14: The precision, recall, and F1 score of LLMDFA and LLM-based end-to-end analysis in the DBZ, XSS, and OSCI detection. From left to right in each line from left to right, the sub-figures depict the statistics of performance in the DBZ, XSS, and OSCI detection, respectively.

### A.3.4 Performance of LLMDFA and it Ablations Using Different LLMs

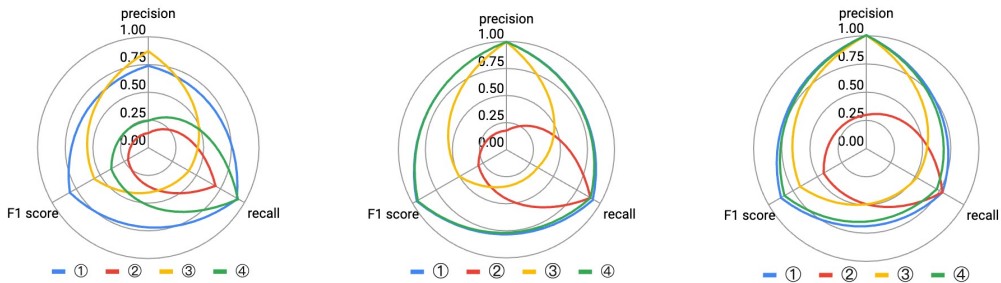

(a) Performance comparison of LLMDFA and ablations powered by gpt-3.5

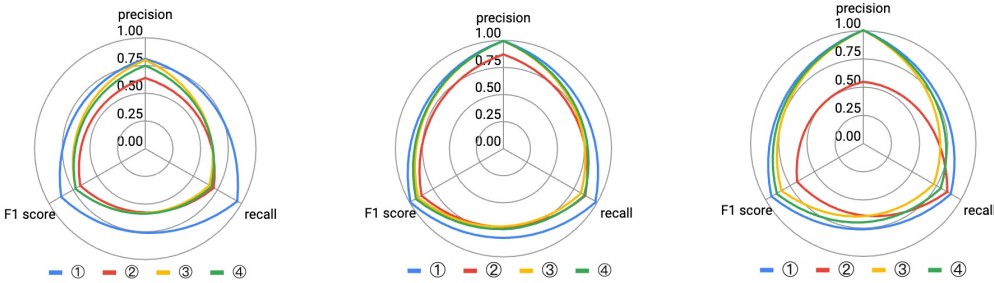

(b) Performance comparison of LLMDFA and ablations powered by gpt-4.0

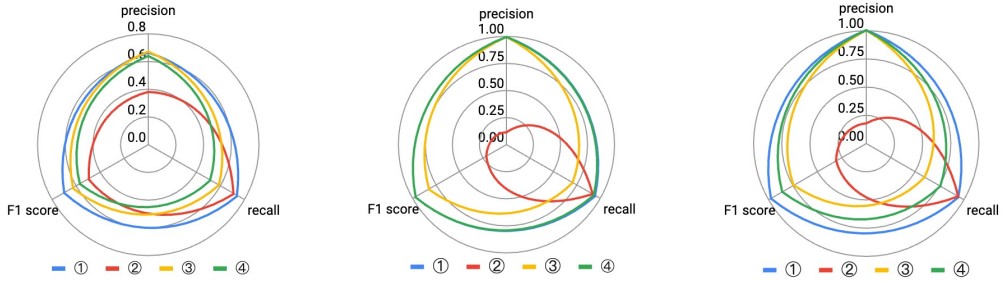

(c) Performance comparison of LLMDFA and ablations powered by gemini-1.0

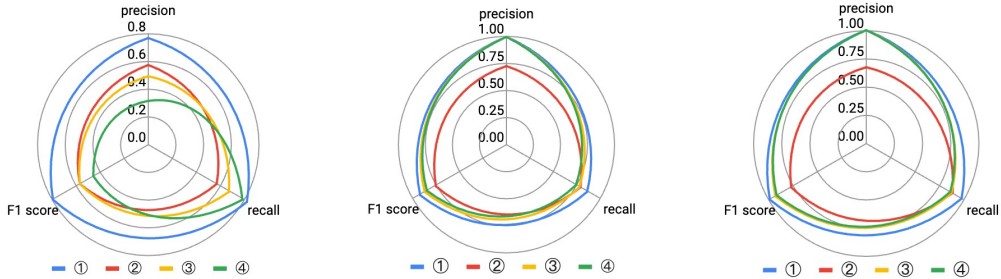

(d) Performance comparison of LLMDFA and ablations powered by claude-3

Figure 15: The performance of LLMDFA(①), NoSynExt(②), NoCoT(③), and NoSynVal(④) using `gpt-3.5`, `gpt-4`, `gemini-1.0`, and `claude-3` in the DBZ, XSS, OSCI detection. From left to right in each line from left to right, the sub-figures depict the statistics of performance in the DBZ, XSS, and OSCI detection, respectively.

### A.4 Case Study

This section presents several examples of hallucinations, including the ones in the LLM-based end-to-end analysis and the incorrect path conditions synthesized by LLMDFA in the path feasibility validation. We also provide examples of false positives and false negatives reported by LLMDFA when analyzing real-world Android malware applications in TaintBench.

#### A.4.1 Hallucinations of LLM-based End-to-End Analysis and Ablations

As demonstrated in the main body of our paper, LLMDFA mitigates the hallucinations of LLM-powered dataflow analysis according to two key ideas. First, it decomposes the whole analysis into three more manageable sub-problems, which target smaller code snippets and more simple program properties. Second, it leverages the tool synthesis and the few-shot CoT prompting to solve the three sub-problems effectively. In what follows, we present three typical cases of hallucinations that the LLM-based end-to-end analysis and the ablations of LLMDFA can suffer.

**Case I:** The LLM-based end-to-end analysis and the ablation NoSynExt without extractor synthesis would identify Integer.MIN_VALUE as a potential zero value, which does not conform to Java semantics. In contrast, LLMDFA correctly identifies sources and sinks in a deterministic fashion with the extractors, which are synthesized by LLMs according to user-specified examples.

Listing 2: An example of misidentified sources

```
int data;
data = Integer.MIN_VALUE;
 Read data from cookies
Cookie cookieSources[] = request.getCookies();
if (cookieSources != null) {
   POTENTIAL FLAW: Read data from the first cookie value
  String stringNumber = cookieSources[0].getValue();
  try {
    data = Integer.parseInt(stringNumber.trim());
  } catch(NumberFormatException exceptNumberFormat){
    IO.logger.log(Level.WARNING, "Number format exception", exceptNumberFormat);
  }
}
badSink(data , request, response);
```

**Case II:** The ablation NoCoT would identify a dataflow fact from the zero value to the first assignment to dataContainer.containerOne, which does not conform to the control flow order. The LLM-based end-to-end analysis is instantiated with the few-shot CoT prompting strategy. However, it takes the whole program as the input and, thus, may suffer more severe hallucinations due to the lengthy prompts, making it identify the incorrect dataflow facts in Listing 3.

Listing 3: An example of misidentified dataflow facts in single functions

```
int data = 2;
Container dataContainer = new Container();
dataContainer.containerOne = data;
goodG2BSink(dataContainer, request, response);
data = 0;
dataContainer.containerOne = data;
badSink(dataContainer, request, response);
```

**Case III:** The LLM-based end-to-end analysis and the ablation NoSynVal (i.e., the ablation that directly prompts LLMs for path feasibility validation) would regard the condition in the second if-statement as a satisfiable one when data has a zero value, causing a false positive in this case.

Listing 4: An example of misidentified feasible paths

```
if (IO.STATIC_FINAL_TRUE) {
  data = 0.0f;
} else {
  data = 2.0f;
}
if (Math.abs(data) > 0.000001) {
  int result = (int)(100.0 / data);
  IO.writeLine(result);
}
```

In our evaluation, we have shown that LLMDFA achieves better performance than the LLM-based end-to-end analysis and its three ablations. The above three typical cases of hallucinations can be effectively mitigated by LLMDFA. The statistics and these concrete cases demonstrate the effectiveness of our problem decomposition and the technical designs of LLMDFA, including the tool synthesis and few-shot CoT prompting.

### A.4.2 Incorrect Path Conditions in Path Feasibility Validation

We present several examples to show the limitations of LLMDFA in path feasibility validation. Specifically, we discuss three complex forms of path conditions.

**Case I: Usage of Library Functions.**

Listing 5 shows the code snippet in CWE369_DBZ__float_connect_tcp_divide_09.java. The condition of the second if-statement is `Math.abs(data)>0.000001`, which is apparently not satisfied when `data` is equal to 0. In our evaluation, we find that LLMDFA tends to take `Math.abs(data)` as a constraint and append it to a Z3 instance directly, which yields a crash in the execution of the synthesized script. After several rounds of fixing, LLMDFA can generate a script program encoding the path condition correctly. However, we still observe that LLMDFA can fail to synthesize the correct script programs for several benchmark programs. Specifically, LLMDFA can wrongly interpret the semantics of `Math.abs(data)>0.000001` with the conjunction `And(data > 0.000001, data < -0.000001)`, while the correct interpretation should be the disjunction `Or(data > 0.000001, data < -0.000001)`. It shows one of the limitations of LLMDFA. When a branch condition contains a library function, LLMDFA may offer a wrong interpretation of its semantics. Although interpreting `Math.abs(data)>0.000001` as `And(data > 0.000001, data < -0.000001)` also makes LLMDFA identify the path as an infeasible one, the path is not discarded in a correct way.

Listing 5: CWE369_DBZ__float_connect_tcp_divide_09

```
public class CWE369_DBZ__float_connect_tcp_divide_09 {
  public void goodB2G1() {
    if (IO.STATIC_FINAL_TRUE) {
        data = 0.0f;
    } else {
      data = 2.0f;
    }

    if (Math.abs(data) > 0.000001) {
        int result = (int)(100.0 / data);
        IO.writeLine(result);
    }
  }
}
```

**Case II: Usage of User-defined Functions**

Listing 6 shows an example of using a user-defined function in a branch condition. Specifically, the function `staticReturnsTrueOrFalse` randomly generates `True` or `False` as the return value. Actually, the branch condition should be symbolized with an unconstrained variable in a Z3 instance. In our current implementation of LLMDFA, we do not support the retrieval of the function definition of `staticReturnsTrueOrFalse`, and the LLMs may directly interpret the branch condition as `True` or `False` incorrectly. For the program shown in Listing 6, LLMDFA would cause a false negative if the branch condition is interpreted as `False`.

Listing 6: CWE369_DBZ__float_connect_tcp_divide_12

```
public class CWE369_DBZ__float_connect_tcp_divide_12 {
  public void bad() {
    if(IO.staticReturnsTrueOrFalse()) {
        data = 0.0f;
    } else {
      data = 2.0f;
    }

    if (IO.staticReturnsTrueOrFalse()) {
      int result = (int)(100.0 / data);
      IO.writeLine(result);
    }
  }
```

```
    public boolean staticReturnsTrueOrFalse() {
      return (new java.util.Random()).nextBoolean();
    }
  }
```

## Case III: Usage of Global Variables

Listing 7 shows an example where the branch condition is guarded by a static member field `badPublicStatic`, which can be regarded as a global variable for the member functions, such as the functions `bad` and `badSink`. Initially, `badPublicStatic` is set to `False`. The function `bad` further sets it to `True`. Hence, the branch condition `badPublicStatic` is satisfied when the function `badSink` is invoked after the statement `badPublicStatic = true` in the function `bad`.

In our implementation of LLMDFA, we just retrieve the initialization of each class field via simple parsing. LLMDFA can introduce a false negative in this example, as it is not aware that `badPublicStatic` is set to `True`. By simple parsing, we cannot obtain the precise value of a global variable that can be modified at multiple program locations. Once the global variables are used to construct a branch condition, LLMDFA may determine the feasibility of the program path incorrectly, introducing false positives or false negatives. To prune the infeasible path with high recall, we can concentrate on the branch conditions in simple forms. For example, if we encounter a branch condition using a global variable, we can assume that it can be satisfied. Although this strategy may introduce false positives, we still have the opportunity to reject many infeasible paths.

Listing 7: CWE369_DBZ__float_connect_tcp_divide_22a

```
public class CWE369_DBZ__float_connect_tcp_divide_22a {
  public static boolean badPublicStatic = false;

  public void bad() {
    badPublicStatic = true;
    int data = 0;
    badSink(data);
  }

  public void badSink(int data) {
    if (badPublicStatic) {
      int result = (int)(100.0 / data);
      IO.writeLine(result);
    }
  }
}
```

Lastly, it is worth noting that encoding path conditions precisely is quite challenging. Existing studies of compilation-based dataflow analysis have explored it for several decades while still failing to achieve satisfactory performance in real-world scenarios. In our work, LLMDFA has demonstrated its potential in discovering the path conditions that exhibit conflicting patterns. Notably, it can understand commonly used library functions according to their names, while such library functions can not be analyzed by compilation-based approaches when their implementations are unavailable. In the future, we can further improve LLMDFA in encoding path constraints by more advanced strategies, such as adopting the CoT prompting to the library function encoding and feeding LLMs with more detailed path info. Besides, we can enforce LLMDFA only focus on specific simple individual branch conditions, such as `p != 0`, and generate the condition that over-approximates the actual path condition. This strategy can eventually achieve semi-path sensitivity, which would improve the precision of specific analysis instances with little sacrifice of the recall.

### A.4.3 Examples of False Positives and Negatives of LLMDFA upon TaintBench

Listing 8: A false negative example of LLMDFA upon TaintBench

```
private String getSmsMessagesout() {
  String[] projection = new String[]{"id", "address"};
  StringBuilder str = new StringBuilder();
  //uri is a source
  Cursor uri = getContentResolver().query(Uri.parse("content://sms/sent"), projection, "id desc");
  str.append(processResults(uri, true));
  return str.toString();
}

private StringBuilder processResults(Cursor cur, boolean all) {
  //cur: Start point of the missed dataflow summary
  StringBuilder str = new StringBuilder();
  try {
    while (cur.moveToFirst()) {
      ...
      String phoneNumber = cur.getString(phoneColumn);
      str.append(phoneNumber + "\n");
    }
  } catch (Exception e) {
    Log.e(PhoneSyncService.TAG, e.toString());
  }
  ...

  //str: End point of the missed dataflow summary
  return str;
}

public void BackConnTask() {
  Socket socket = new Socket(InetAddress.getByName("www.roidsec.com"), 5001);
  ...
  outStream.write(("result_Messageout" + getSmsMessagesout()).getBytes("GBK")); //The argument is a
      sink
  outStream.write("\n".getBytes());
  ...
}
```

Listing 9: A false positive example of LLMDFA upon TaintBench. The lengthy function *onMessage* makes LLMDFA identify a spurious dataflow summary from *url* to the argument *msg*. The spurious dataflow summary eventually introduces a spurious dataflow path from a source value *url*, i.e., the return value of *getStringExtra*, to a sink value in the function *sendBackgroundMessage*.

```
public void onMessage(Context context, Intent intent) {
  Utils utils = Utils.getInstance(context);
  String command = intent.getStringExtra("command");
  final Context scontext;
  String url;
  ...
  if (command.equals("START")) {
    context.startService(new Intent(context, WorkService.class));
  } else if (command.equals("SEND_SMS")) {
    String msg = intent.getStringExtra("data");
    utils.sendBackgroundMessage(msg); //msg: End point of the spurious dataflow summary
  } else if (command.equals("SEND_SMS_NOW")) {
    String sms_now = intent.getStringExtra("data")
    utils.sendMessageNow(sms_now);
  } else if (command.equals("CHANGE_URL")) {
    url = intent.getStringExtra("data"); //url: Start point of the spurious dataflow summary. It is
        also a source.
    if (url != null && !url.equals(BuildConfig.FLAVOR)) {
      utils.setUrl(url);
    }
  }
  ...
  else if (command.equals("TASK")) {
    scontext = context;
    final String data = intent.getStringExtra("data");
    new Thread(new Runnable() {
      public void run() {
        Utils.getInstance(scontext).installTask(data);
      }
    }).start();
  }
}
```

