# OpenReview forum: "LLMDFA: Analyzing Dataflow in Code with Large Language Models"
_NeurIPS.cc/2024/Conference — NeurIPS 2024 poster_

### Official Review · Reviewer_UBFi · 2024-06-12

**Soundness:** 3
**Presentation:** 2
**Contribution:** 3
**Rating:** 7
**Confidence:** 3

**Summary:**

This paper proposes an LLM-based approach for detecting bugs with data flow analysis. The proposed approach LLMDFA consists of three parts: source/sinks extraction, dataflow summarization, and path feasibility validation. LLMDFA allows LLMs to interact with different program analysis tools. The authors conduct experiments on bug detection in both synthetic and real-world datasets.

**Strengths:**

+The authors proposed an interesting approach for solving an important question in program languages.

+LLMDFA decomposes bug detection into three steps, which are proven useful for complex data flow reasoning.

+Experiments are conducted on both synthetic and real-world datasets.

**Weaknesses:**

-The results on synthetic datasets are very high, maybe the dataset is too easy?

-All components in LLMDFA require multiple rounds of try-and-refine. It is better to discuss the cost of multiple-round generation.

**Questions:**

- Most baselines are non-deep-learning approaches. I wonder if there are any deep learning-based approaches previously proposed for similar bug detection tasks?

- In my understanding, it is possible to write a single parser to extract the sources/sinks for all programs. What's the advantage in using LLM to generate different parsers for different programs? Also, I'd like to see more discussion on the impact of ASTs in source/sink extraction.

- I’m a little confused about how the results of path feasibility validation can lead to the detection of bugs. Can the authors explain in more detail?

**Limitations:**

The limitations have been well-discussed.

---

> ### Author Rebuttal · Authors · 2024-08-06
>
> ### **I. Answers to Questions**
> **Q1. Comparison with ML-based approaches**
>
> Targeting a compilation-free and customizable dataflow analysis, we have conducted a comprehensive survey of existing literature and have not encountered any ML-based approaches that specifically tackle the same problem as our research. The most related work is DeepDFA, which is cited in our paper as [25]. Specifically, DeepDFA introduces abstract dataflow embedding and applies a classification model to the embedding results for bug detection. By jointly training the embedding and classification models upon a large dataset containing targeted bugs, DeepDFA can obtain the capability of detecting the target bugs. Although it enables compilation-free analysis, it requires the training process to be repeated for individual dataflow analysis problems. Also, a large amount of training data may not be available for specific bug types. Hence, the lack of customization support limits the applicability of DeepDFA in real-world scenarios where flexible dataflow analysis requirements are necessary. These differences also prevent us from devising a fair empirical comparison between LLMDFA and DeepDFA.
>
> In our evaluation, we hence choose few-shot CoT prompting as an LLM-based end-to-end solution that has the same assumptions as our technique. We acknowledge the Reviewer UBFi's suggestion to include further discussions on existing machine learning-based approaches in our paper. We will further add more justifications for the baseline choices in the evaluation section and compare LLMDFA with existing ML-based analysis from a methodological standpoint.
>
> **Q2. More details of source/sink extractor synthesis**
>
> First, as demonstrated in lines 169 and 170 in Section 3.1, we synthesize source/sink extractors for each bug type and apply them to different programs instead of synthesizing extractors for individual programs. If we target a specific bug type, the source/sink extractor synthesis is a one-time effort. The synthesized extractors can be deterministically applied to the target programs without introducing any hallucinations.
>
> Second, manually specifying extractors requires the users to exhibit expert knowledge of compilation infrastructures, such as the tree-sitter parsing framework. Considering the tree-sitter-java supporting the parsing of Java programs, for example, 528 node types correspond to various Java constructs, such as call expressions and method declarations. Besides, these nodes exhibit diverse sets of child nodes. The intricacy of the parsing library poses a challenge for non-experts in manually creating extractors to cater to different analysis requirements.
>
> Third, the impact of source/sink extractor synthesis is demonstrated by our ablation study in Section 4.4. As shown in Figure 8, if we identify sources and sinks with LLMs directly instead of using synthesized extractors, the ablation, namely NoSynExt, yields a much lower precision, recall, and F1 score than full-featured LLMDFA. This indicates that the extractor synthesis enables us to mitigate the hallucinations in the source/sink extraction and eventually significantly improves the performance of the overall analysis. We also offer a concrete example in Appendix A.4.1 as a case study (See Case I) to show the spurious sources identified by LLMs while avoided by the synthesized extractors.
>
> We appreciate Reviewer UBFi's comments on the source/sink extractor synthesis. We will include more discussions in our paper.
>
> **Q3. More details of path feasibility validation**
>
> The path feasibility is a necessary condition of a dataflow path that may introduce a bug. As shown by the motivating example in Figure 1, the value of x at line 9 can be propagated to the argument z at line 13 and then propagated to the second parameter in the function foo. Furthermore, it is utilized at line 4 as a divisor. Such a dataflow path can be derived by concatenating the summaries of the two functions. However, it does not imply the bug’s existence, as the branch condition at line 3 is violated when the value is equal to 0. This means that we have to check the path feasibility after we concatenate the dataflow summaries obtained in the second stage. We will motivate the path feasibility validation with more explanations.
>
> ### **II. Other Concerns**
> **C1. Dataset Details**
>
> Juliet Test Suite is a synthetic benchmark widely used for evaluating static analysis tools. The programs in the benchmark are relatively small. That said, it should be noted that the programs in Juliet Test Suite still exhibit complex features, such as complex path conditions, library function usage, and data structure usage. On the other hand, the programs in TaintBench are all real-world Android malware applications. Therefore, the high performance upon Juliet Test Suite and promising results upon TaintBench demonstrate the potential of LLMDFA in discovering dataflow facts for bug detection.
>
> During the response period, we further extended LLMDFA to support C/C++ and JavaScript analysis and meanwhile evaluated it upon three bug tracks in Juliet Test Suite for C/C++ and SecBench.js [R2]. The latter contains 138 bugs in real-world JavaScript programs, including command injection, taint path, and code injection vulnerabilities. Notably, many of the vulnerabilities have been assigned with CVE IDs due to their significant security impact. The results are shown in Table 9 and Table 10.
>
> The above datasets cover different program domains, programming languages, and bug types. As demonstrated by empirical evaluation results, the satisfactory performance of LLMDFA can show its potential value in analyzing real-world programs in a compilation-free and customizable manner.
>
>
> **C2. Cost of LLMDFA**
>
> Please refer to the common concern in the global response.

---

### Official Review · Reviewer_ep1y · 2024-06-17

**Soundness:** 2
**Presentation:** 3
**Contribution:** 3
**Rating:** 6
**Confidence:** 4

**Summary:**

The paper proposes LLMDFA, a compilation-free and customizable dataflow analysis, which takes a program and its CFG as input.
The authors decompose the problem into several subtasks including Source/Sink Extraction, Dataflow Summarization, and Path feasibility Validation, and introduce some strategies. They evaluate LLMDFA on synthetic programs to detect three representative types of bugs and on real-world Android applications for customized bug detection.

**Strengths:**

+ easy to follow
+ interesting idea
+ impressive improvement

The presentation is easy to follow and the experiment results are impressive.

**Weaknesses:**

- some designs are not well-motivated
- high cost

I have some reservations about the efficacy and meaningfulness of employing Z3 for verification, especially considering the preceding processes are generated by LLMs, which might inherently carry some inaccuracies. It seems challenging to ascertain whether the results from the final Z3 verification justify the associated costs, given these potential uncertainties. When Z3 fails to verify, what is the possible fixing mechanism? How to locate the error?

Additionally, preparing code for Z3 verification involves intricate steps such as defining data types, data structures, and functions, and then also need to translate constraints into Z3's language. While I appreciate that the authors admit these challenges with illustrative examples shown in the appendix, the results presented in Table 1 are notably impressive. Could this suggest that the dataset primarily comprises simpler structures and operations? It would be enlightening if more detailed results for the specific tasks mentioned could be shared, beyond a general ablation study.

Finally, comparing the costs of traditional methods and those involving LLMs directly can be quite complex. Including a more nuanced comparison of human effort and financial implications might provide clearer support for the strengths and contributions of the LLMDFA approach.

**Questions:**

- What is the cost of each part?
- How much validation code using Z3 can be successfully generated?
- Can the whole procedure be automated? Or how can the user be involved in LLMDFA?

**Limitations:**

The limitation has been discussed.

---

> ### Author Rebuttal · Authors · 2024-08-05
>
> ### **Answers to Questions**
>
> **Q1. The costs of the three stages**
>
> Please refer to the common concern in the global response.
>
> **Q2. The statistics and details of path feasibility validation**
>
> In Appendix A.3.2, we demonstrate the detailed statistics of path feasibility validation. In our evaluation, we set the upper bound of the fixing number to three in the script program synthesis. According to Figure 13 (a), 75.20%, 96.43%, and 85.25% of scripts are synthesized using gpt-3.5 without any fixes in the DBZ, XSS, and OSCI detection. After feeding the error messages thrown by the Python interpreter iteratively, 99.39%, 96.43%, and 95.08% script programs can be synthesized within three fixes, respectively. For the remaining cases, which only take up 0.61%, 3.57%, and 4.92%, we leverage LLMs to directly determine whether the path is feasible or not. Similar observations can also be derived from the results when using other LLMs.
>
> We also noticed that not all successfully synthesized scripts encode the path conditions correctly. We have shown three typical examples in Appendix A.4.2. We further investigated all the synthesized cases and found that 78.57%, 88.68%, and 83.65% of synthesized scripts encode the path conditions correctly. As pointed out by the reviewer, complex operations, such as library functions, can cause LLMs to fail to generate precise path conditions.
>
> Lastly, it is worth mentioning that partially correct path encoding may be sufficient to filter out false positive bug reports. For example, the dataflow path for a false positive DBZ bug may not have all the path conditions correctly encoded. However, as long as the guard that checks the divisor not being zero is correctly encoded, Z3 can already rule out the bug report. Several existing studies have also reported and utilized similar observations [R12, R13].
>
> **Q3. User Interaction and financial cost**
>
> As shown by the workflow in Figure 4, LLMDFA requires the users to provide a bug specification describing the forms of sources and sinks, few-shot examples containing sources and sinks, and ASTs generated by parsers. Then, the whole workflow can be fully automated without any human intervention. Specifically, the user can specify the bug specification and examples with little effort. In DBZ detection, for example, the user only needs to write six lines of the bug specification and five code examples. We asked five undergraduate students who had a background in Java programming to construct the prompts and found that their average time costs were only around five minutes.
>
> In contrast, customizing the bug detection in a traditional compilation-based static analyzer often involves implementations with hundreds and even thousands of lines of code, which requires expert knowledge of compiler infrastructures, such as the LLVM infrastructure. The lack of customization support prevents the wide adoption of existing bug detectors, which is also stated by existing studies [R14].
>
> We also measure the token costs of LLMDFA and further compute its financial costs of detecting different bug types. For more details, please refer to the response to the common concern in the global response.

---

> > ### Comment · Reviewer_ep1y · 2024-08-09
> >
> > Thank you for your time.
> > I think the LLM's hallucination reduces the effectiveness of the verification Z3 in the final stage and may add some cost (as other reviewers agree) since most parts are generated by the LLM.
> >
> > However, the extensive experimental data and examples you provide demonstrate the potential usefulness of your approach.
> > To further enhance the manuscript, I recommend the following:
> >
> > (1) Clearly delineate the outputs of the LLM versus those elements handled by formal, rigorous methods and show the cost.
> >
> > (2) Provide a detailed, step-by-step running example to illustrate the complex integration process.

---

> > > ### Author Response · Authors · 2024-08-09
> > > **Response to Official Comment by Reviewer ep1y**
> > >
> > > We greatly appreciate your active feedback. Your comments are very important to us in refining the manuscript.
> > >
> > > As stated in the introduction, a key contribution of our work is to reduce the hallucinations of LLMs by applying formal methods. Indeed, the hallucinations of LLMs may reduce the effectiveness of the path feasibility validation with Z3. However, our approach can significantly mitigate the hallucinations of the overall analysis, which has been demonstrated by the superiority of LLMDFA over NoSynVal (See Figure 8) and the satisfactory performance of LLMDFA in bug detection (See Table 1).
> > >
> > > We will revise our manuscript by following your comments.
> > >
> > > - We will enhance the workflow diagram in Figure 4 to annotate the outputs of LLMs and the ones handled by formal methods. Specifically, the extractors in Phase I are synthesized by LLMs, and the dataflow summaries are generated by LLMs in Phase II. In Phase III, i.e., the path feasibility validation, the script encoding the path condition is generated by LLMs, while the script itself invokes a formal method, i.e., SMT solver Z3 for the validation. In the uploaded PDF file, we demonstrate the corresponding token costs (shown in Tables 5 and 6) and average financial cost of each phase (shown in Tables 7 and 8). We will include them in the evaluation or Appendix and add the necessary explanations.
> > >
> > > - At the ends of Sections 3.1, 3.2, and 3.3, we had a step-by-step running example at the end of each section. Also, the listings in the supplementary material offer the details of prompts and responses. Particularly, Listings 6 and 7 in the supplementary material show the iterative process of path condition encoding. We will append the prompts to Appendix and add pointers in the main-text pointing to the individual prompt examples.
> > >
> > >
> > > Please let us know if we understand your questions correctly and if our answers are satisfactory.

---

> > > > ### Comment · Reviewer_ep1y · 2024-08-11
> > > >
> > > > Thank you for the comprehensive clarifications provided in response to the concerns raised. Your answers have addressed the major points effectively, and I appreciate the effort to explain and justify the methodologies used.
> > > >
> > > > One remaining suggestion is the example. The current example, while effective in showcasing the capabilities of your approach, lacks a degree of naturalness and seems designed to emphasize complexity rather than clarity. This choice might obscure the real-world applicability and accessibility of your research to a broader audience.
> > > > I would recommend selecting a simpler, real-world example that clearly illustrates how your techniques can be applied in practical scenarios.

---

> > > > > ### Author Response · Authors · 2024-08-11
> > > > > **Response to Official Comment by Reviewer ep1y**
> > > > >
> > > > > Thank you for your feedback. We will follow your suggestion and use a simpler example to demonstrate DBZ bugs. For example, we consider simplifying the branch condition to `b != 0` directly. More complex cases, such as the ones containing library function calls in branch conditions, will be appended as case studies in Appendix. It would make the example more intuitive and accessible for a broader audience.
> > > > >
> > > > > Thank you again for your time and effort during the review and rebuttal.

---

### Official Review · Reviewer_jfEP · 2024-07-07

**Soundness:** 2
**Presentation:** 3
**Contribution:** 3
**Rating:** 6
**Confidence:** 4

**Summary:**

This paper presents LLMDFA, an LLM-powered compilation-free and customizable dataflow analysis framework for bug detection. To mitigate LLM hallucination, LLMDFA breaks down the dataflow analysis (DFA) task into multiple subtasks and delegates complex reasoning to specialized external tools. In bug detection evaluations involving synthetic programs and real-world Android applications, LLMDFA achieves a precision of 87.10% and a recall of 80.77% on average. These results outperform existing baselines.

**Strengths:**

1. This paper investigates a promising direction in leveraging LLMs for code dataflow analysis. The motivation and methodology are clearly articulated and well-structured.
2. To mitigate LLM hallucinations, the authors introduce a novel three-stage workflow: In the Source/Sink Extraction and Path Feasibility Validation stages, LLMDFA employs an LLM as a coordinator, delegating the complex analysis to external tools. In the Dataflow Summarization stage, LLMDFA uses a few-shot chain-of-thought (CoT) prompting strategy to align LLMs with program semantics. These approaches effectively reduce the intrinsic hallucinations associated with LLMs.
3. The empirical results demonstrate that LLMDFA significantly outperforms existing baselines, including the classic dataflow analyzer CodeFuseQuery and an end-to-end CoT solution.
4. LLMDFA can be applied to arbitrary programs without the need for compilation.

**Weaknesses:**

1. In the Path Feasibility Validation stage, the LLM's role is to synthesize a Z3 solver script. However, producing a runnable Z3 solver script does not necessarily ensure that the path is feasible, as the generated Z3 constraints may not accurately reflect the actual path conditions.
2. Since LLMDFA does not require compilation, the presence of any unknown external call (e.g., `a = unknown_func(a)`) can terminate the dataflow analysis. This scenario is particularly common in real-world code.

**Questions:**

Based on my best understanding, the dataflow summarization stage generates dataflow paths between sources and sinks. I am curious if any deterministic algorithms could handle this task instead of relying on an LLM. Furthermore, considering the potentially numerous rules involved in forming the paths, could you provide the actual prompt used at this stage?

**Limitations:**

The authors addressed limitations in their work.

---

> ### Author Rebuttal · Authors · 2024-08-05
>
> ### **I. Answers to Questions**
> **Q1. Technical design of dataflow summarization**
>
> Many existing techniques, such as Pinpoint [R10] and SVF [R11], can derive dataflow paths deterministically. However, they all require the intermediate representations generated by a successful program compilation. As motivated by Figure 2, we want to fill the gap and establish a compilation-free program analysis. That is why we do not apply existing deterministic techniques and instead leverage LLMs for code semantic interpretation to achieve dataflow summarization.
>
> We uploaded the prompts and responses as supplementary material. The details of dataflow summarization can be found in Listing 5, where the prompt and response of this phase are clearly demonstrated. By specifying a few typical patterns of value propagation, such as assignments and direct uses, we can inspire the LLMs to discover dataflow paths through the few-shot CoT prompting. This process does not require numerous rules specifying the dataflow facts induced by various program constructs, as the LLMs can capture the essence of the relationship between two values due to their exceptional generalization ability in code understanding.
>
> ### **II. Other Concerns**
> **C1.Hallucinations of Path Feasibility Validation**
>
> Although the synthesized Z3 scripts may not correctly encode path conditions, we find that 78.57%, 88.68%, and 83.65% of them are correct in the DBZ, XSS, and OSCI detection, respectively. The most common reason for incorrect encoding is complex operations that appear in branch conditions, such as library functions.
>
> Another point we want to make is that partially correct path encoding may be sufficient to filter out false positive bug reports. For example, the dataflow path for a false positive DBZ bug may not have all the path conditions correctly encoded. However, as long as the guard that checks the divisor not being zero is correctly encoded, Z3 can already rule out the bug report.  Several existing studies have also reported and utilized similar observations [R12, R13].
>
> **C2.External Function Call**
>
> Handling external function calls is a classic challenge in static analysis. In our work, LLMDFA validates a dataflow path based on the summaries generated in the second phase. Without the summaries of external functions, it can still discover dataflow paths formed by internal function calls (i.e., the functions available in the code base).
>
> Besides, for commonly used external function calls, such as standard library functions, LLMDFA can potentially understand the dataflow facts induced by their invocations. For example, when analyzing the project `samsapo` in TaintBench, LLMDFA encounters the following code snippet where the function `append` is a library function of the class `StringBuilder`. Due to the lack of its function body in the prompt, LLMDFA cannot summarize the dataflow facts for the callee function `append`. However, it can derive the dataflow summary from `str` to `stringBuilder`, which indicates that the value of the parameter `str` can propagate to the StringBuilder variable. This benefits from the ability of LLMs in code understanding, especially for the commonly used library functions.
>
> ```
> public static Fragment instantiate(Context context, String str, Bundle bundle) {
>     ...
>     stringBuilder = stringBuilder.append(str);
>     ...
> }
> ```

---

> > ### Comment · Reviewer_jfEP · 2024-08-12
> > **Thank you for your response**
> >
> > Dear authors,
> >
> > Thank you for your diligent efforts and comprehensive responses. Your explanations have effectively addressed my concerns. As a result, I am pleased to elevate my confidence in your work from 3 to 4 and remain cautiously optimistic about your paper.

---

> > > ### Author Response · Authors · 2024-08-12
> > >
> > > Thank you for your feedback and recognition. We will carefully revise the corresponding presentation issues as per your suggestions in the review, and supplement additional experimental data. Thank you again for your efforts during the review and rebuttal process.

---

### Official Review · Reviewer_KJcQ · 2024-07-10

**Soundness:** 2
**Presentation:** 2
**Contribution:** 2
**Rating:** 5
**Confidence:** 3

**Summary:**

The author proposes a data flow analysis tool based on LLM, named LLMDFA. This tool addresses the LLM hallucination problem by decomposing tasks. The method is divided into three stages: the first stage involves calling and extracting sources and links through LLM-generated scripts; the second stage generates summaries of the data flow; the third stage conducts path analysis using synthesis tools and scripts. The study was validated on the Java programming language and considered four closed-source LLM models. Experimental results show that LLMDFA has high Recall and F1 metrics.

**Strengths:**

- The author explores the effective application of LLM in the field of code analysis.
- Utilizes the generative capabilities of LLM to call external programs.

**Weaknesses:**

- The third part is relatively obscure. Although the author attempts to formalize the method of LLM analysis data flow, this expression is difficult for readers lacking a background in code analysis and data flow analysis to understand.
- This paper is an application-type article of LLM. The generalizability of the research has not been fully verified, as experiments were only conducted on the Java language, limiting the general applicability of the conclusions.
- LLMDFA is still affected by the LLM hallucination problem and may generate incorrect summaries, but the experimental section does not explore the impact of this issue on the method.
- The experimental data volume is small, with only 37 programs tested for each type of bug, which may lead to significant bias in the experimental results. It is recommended to verify with a larger dataset to improve the reliability of the conclusions. TaintBench also has only 39 real-world android malware (line 242). The author should verify on a larger dataset to ensure the feasibility of the conclusions. Some datasets can be referenced [1,2].
- The related work section lacks an introduction to the current research progress of LLM in the field of code analysis, such as [3].

**Citation Suggestions:**
[1] Yiu Wai Chow, Max Schäfer, Michael Pradel. 2023. "Beware of the Unexpected: Bimodal Taint Analysis." *Proceedings of the 32nd ACM SIGSOFT International Symposium on Software Testing and Analysis (ISSTA 2023)*. ACM, New York, NY, USA, 211–222. https://doi.org/10.1145/3597926.3598050
[2] Liu Wang, Haoyu Wang, Ren He, Ran Tao, Guozhu Meng, Xiapu Luo, Xuanzhe Liu. 2022. "MalRadar: Demystifying Android Malware in the New Era." *Proc. ACM Meas. Anal. Comput. Syst.* 6, 2, Article 40 (June 2022), 27 pages. https://doi.org/10.1145/3530906
[3] Haonan Li, Yu Hao, Yizhuo Zhai, Zhiyun Qian. 2024. "Enhancing Static Analysis for Practical Bug Detection: An LLM-Integrated Approach." *Proc. ACM Program. Lang.* 8, OOPSLA1, Article 111 (April 2024), 26 pages. https://doi.org/10.1145/3649828

**Questions:**

- Your method has so far only been validated on the Java language. Do you have plans to experiment with other programming languages such as Python, C++, etc.? If so, what are the experimental results?
- How do you think LLMDFA applies to other programming languages? Does the method need to be adjusted to adapt to different programming environments?
- Why not try using open-source models?

**Limitations:**

See Weaknesses

---

> ### Author Rebuttal · Authors · 2024-08-05
>
> ### **I. Answers to Questions**
> **Q1. Experiments upon Other Languages**
>
> Although we did not claim any contribution of multi-lingual support, it is easy to extend LLMDFA to support other languages due to its compilation-free design. During the rebuttal, we extended it to support C/C++/JavaScript and evaluated it with gpt-3.5 upon the Juliet Test Suite for C/C++ and SecBench.js benchmark [R2]. Due to the limited time, we did not evaluate LLMDFA using the other three LLMs.
>
> The Juliet Test Suite for C/C++ does not contain XSS bugs. Hence, we chose another bug type, namely Absolute Path Traversal (APT), which is a typical security vulnerability in Common Weakness Enumeration. As shown in Table 9, the performance of LLMDFA on C/C++ is equally good as on Java.
>
> SecBench.js includes 138 vulnerabilities in JavaScript packages, covering command injection, taint path, and code injection vulnerabilities, which have been assigned with CVE IDs due to significant security impact. Table 10 shows that our compilation-free analysis upon SecBench.js achieves 92.54% precision and 71.74% recall, which are comparable with the compilation-dependent approach in [R2].
>
> **Q2. Migration Details for Other Languages**
>
> LLMDFA can be extended to support other languages with low implementation effort. Based on dataflow analysis theory, we only need to construct the control flow graph with corresponding tree-sitter packages and then reuse the current implementation. To migrate Java analysis to C/C++/JavaScript analysis, we changed no more than 100 lines of code. Most of the changes relate to the changes to AST node types and parser initialization.
>
> **Q3. Reasons for Not Using Open-Sourced Models**
>
> We actually tried to utilize open-sourced models, such as CodeLlama-7b-Instruct-hf. However, these models do not seem to have the capabilities to meet the output constraints in the presence of lengthy prompts. In several rounds of prompting for extractor synthesis, the open-sourced models, such as CodeLlama-7b-Instruct-hf, kept generating the following response without synthesizing the extractors at all.
>
> ```
> I will examine the examples and their abstract syntax trees according to the given instructions. I will then generate the extractors to identify desired sources and sinks. Let me start the synthesis.
> ```
>
> Similarly, the LLMs are required to generate Yes or No to indicate whether the value of a program variable can depend on the value of another value. However, CodeLlama-7b-Instruct-hf did not explicitly give Yes or No as its answer in several cases. Instead, it even responded, “It is unknown whether the two values are dependent or not.”
>
> Due to the above reasons, we could not specify a fixed grammar to parse the responses. Similar problems were reported in existing works [R3, R4]. Another option is to use ultra-large open-sourced models. However, it requires GPU resources that are beyond our reach. To mitigate the threat of validity of relying on a single closed-source model, we tried to use as many such models as we could.
>
> ### **II. Other Concerns**
> **C1. Hallucinations in Dataflow Summarization**
>
> The rows named **Summarize** in Table 1 show the precision, recall, and F1 score of dataflow summarization. Consider the DBZ detection using gpt-3.5 as an example. 9.05% of identified dataflow summaries are spurious due to the LLM hallucination in the dataflow summarization. Meanwhile, 2.43% of valid dataflow summaries are not identified by the LLMs. In Appendix A.4.3, we also demonstrate the cases of false positives and negatives caused by the hallucinations in dataflow summarization.
>
> **C2. Evaluation Subjects**
>
> Juliet Test Suite contains 1,850 DBZ, 666 XSS, and 444 OSCI bugs, which are scattered in 3,809, 1,084, and 902 Java files, respectively. The precision, recall, and F1 score of the bug detection are measured upon all these Java files. To measure the performance of the dataflow summarization and path feasibility validation, we have to manually examine the intermediate results. This process would be quite laborious if we checked the intermediate results for all the programs. Hence, we chose 37 programs as the representatives, as the rest of the programs only differ from the selected ones in terms of sources and sinks. We have stated such settings from line 256 to line 259. We will add more justifications to avoid potential confusion.
>
> TaintBench offers rigorous definitions of malicious dataflows, which enable us to write bug specifications and few-shot examples for customization.The bugs in TaintBench are manifested by 53 different source-sink pairs, covering typical buggy patterns of sensitive information leakage. Due to its diverse taint flow forms, TaintBench has been widely used by many research papers published in top-tier conferences in software engineering and security [R5, R6, R7]. We believe choosing this benchmark can provide strong empirical evidence of LLMDFA’s effectiveness in the real world.
>
> We appreciate the reviewer's recommendation of the two works [R2, R8]. However, the benchmarks they used are not suitable for the evaluation during the rebuttal. First, the constructed benchmark in [R8], namely MalRadar, and all the datasets listed by Table 1 in [R8], are currently unavailable or require authentication. Second, the experimental subjects used in [R2] include proprietary projects on LGTM.com and an open-sourced real-world benchmark SecBench.js. We tried our best to extend and evaluate LLMDFA upon SecBench.js during the rebuttal phase. For more evaluation details upon SecBench.js, please also refer to the answer to Q1 as above.
>
> **C3. Related Work**
>
> Actually, we cited the preprint of the work LLift [R9] as [47]. By inferring how library functions initialize variables with LLMs, LLift detects uninitialized variables in the program. Different from LLMDFA, it still requires a compilation-dependent analyzer as its backend and can only detect uninitialized variables without customization support.

---

> > ### Comment · Reviewer_KJcQ · 2024-08-12
> > **Reply to rebuttal by authors**
> >
> > Thank you for your efforts to make a comprehensive response. Your explanations have addressed my concerns. I am happy to improve my scores.

---

> > > ### Author Response · Authors · 2024-08-12
> > >
> > > Thank you for your recognition of our rebuttal content. We will add a discussion on multi-language support and other necessary experimental data in the manuscript.

---

### Author Rebuttal · Authors · 2024-08-06

We express our sincere gratitude to all the reviewers for their invaluable feedback and insightful questions. Prior to addressing each individual question and concern raised in the reviews, we present more experimental data, list several additional references, and address the common concerns of reviewers in the global response.

### **I. Additional Experimental Data**

We expanded the experiments during the rebuttal phase and uploaded a PDF file containing six tables as the attachment of this global response. To distinguish the tables from **Tables 1~4** in our original submission, the six tables are labeled with **Tables 5~10**. We will explain the additional experimental results in the individual responses.

### **II. Additional Reference**

In the responses, we will discuss the following papers, some of which are recommended by the reviewers.

[R1] Xia, Chunqiu Steven, et al. "Keep the Conversation Going: Fixing 162 out of 337 bugs for $0.42 each using ChatGPT", ISSTA 2024

[R2] Yiu Wai Chow, et al. "Beware of the Unexpected: Bimodal Taint Analysis", ISSTA 2023

[R3] Beurer-Kellner, et.al. "Prompting is programming: A query language for large language models”, PLDI 2023

[R4] Beurer-Kellner, Luca, et al. "Prompt Sketching for Large Language Models", ICML 2024.

[R5] Li, Kaixuan, et al. "Comparison and Evaluation on Static Application Security Testing (SAST) Tools for Java", FSE 2023.

[R6] Mordahl, Austin. "Automatic testing and benchmarking for configurable static analysis tools", ISSTA 2023.

[R7] Ami, Amit Seal, et al. False negative–that one is going to kill you: Understanding Industry Perspectives of Static Analysis based Security Testing", IEEE S&P 2024.

[R8] Liu Wang, et al. "MalRadar: Demystifying Android Malware in the New Era", Proc. ACM Meas. Anal. Comput. Syst. (June 2022)

[R9] Li, Haonan, et al. "Enhancing Static Analysis for Practical Bug Detection: An LLM-Integrated Approach", OOPSLA 2024.

[R10] Shi, Qingkai, et al. "Pinpoint: Fast and precise sparse value flow analysis for million lines of code", PLDI 2018.

[R11] Sui, Yulei, et al. "SVF: interprocedural static value-flow analysis in LLVM", CC 2016.

[R12] Cheng, Xiao, et al. "Fast Graph Simplification for Path-Sensitive Typestate Analysis through Tempo-Spatial Multi-Point Slicing", FSE 2024.

[R13] Das, Manuvir, et al. "ESP: Path-sensitive program verification in polynomial time", PLDI 2002.

[R14] Johnson, Brittany, et al. "Why don't software developers use static analysis tools to find bugs?", ICSE 2013

### **III. Common Concern**

Now, we address the common concerns of reviewers.

**C1. The costs of the three stages (Reviewer ep1y and Reviewer UBFi)**

We measure the token cost of LLMDFA when analyzing each test program in Juliet Test Suite with different LLMs. In the following, we present the detailed statistics of the token cost of LLMDFA using gpt-3.5, including the input and output token costs of the three phases. The token costs when using the other three models are close to the following statistics.

**Cost of extractor synthesis:**

In the evaluation of extractor synthesis for each bug type, we repeated the experiment twenty times and measured the input and output costs in each round. As shown in Table 5, more than 90% of source extractors can be synthesized with no more than 4,000 input tokens except for the sink extractor synthesis for the OSCI bug detection. Besides, LLMDFA has to iteratively fix the synthesized scripts eleven times in an extreme case when synthesizing the source extractor for the DBZ detection, which introduces a relatively high token cost. However, such extreme cases do not frequently happen in our evaluation. The sub-column $p_{0.9}$ of the column *Input Token Cost* demonstrates the low input token cost of LLMDFA in most cases. We can derive similar observations on the output token cost from Table 5. According to the pricing policy of OpenAI, we obtain the average financial cost of synthesizing source/sink extractor for each bug type in Table 7.

**Cost of dataflow summarization and path feasibility validation:**

Table 6 shows the per-test input/output token costs in the dataflow summarization and path feasibility validation, respectively. Specifically, the token costs are higher in the DBZ detection than the ones in the XSS and OSCI detection. The reason is that the programs containing XSS and OSCI bugs have fewer sources and sinks than the ones containing DBZ bugs. Hence, LLMDFA prompts the LLM fewer times in the XSS and OSCI detection. Similar to the extractor synthesis, we can find that LLMDFA consumes more tokens in a small proportion of cases than the others. In 90% of cases, the input and output token costs of the dataflow summarization do not exceed 24,000 and 800, respectively, while input and output token costs of the path feasibility validation do not exceed 10,000 and 3,000, respectively. Notably, the input token costs in the dataflow summarization are mainly caused by the examples with explanations used for few-shot CoT. Also, the input token costs in the path feasibility validation are mainly introduced by the description of path information and hints for path constraint encoding. Finally, we compute the average per-test financial cost of the last two phases, which are shown in Table 8.


**Overall cost:**

By adding up the costs shown in Table 7 and Table 8, the average financial cost of the DBZ, XSS, and OSCI detection is 0.14 USD, 0.05 USD, and 0.04 USD, respectively. Such a cost is in line with works of a similar nature, such as [R1], which takes 0.42 USD to repair a bug. Notably, the extractor synthesis is one-time for a given bug type. Hence, the financial cost of the detection in practice is even lower.

In our implementation, LLMDFA invokes the online models via OpenAI APIs. The time cost of the LLM inference can be affected by many factors, such as network status, server scheduling, and runtime traffic. Hence, we do not measure LLMDFA's time overhead.

---

> ### Author Response · Authors · 2024-08-11
> **Seek feedback on responses**
>
> We would like to express our sincere gratitude to all the reviewers for the time and efforts they have devoted during the review and rebuttal stages. In our previous global response and individual responses to each review, we have addressed the main concerns regarding this work. In particular, we have answered every question raised in the reviews. As the interactive rebuttal phase is nearing its end, we humbly request that you provide us with some feedback on our previous responses. If any further clarification or explanation of technical details or experimental data is needed, we will respond and provide it promptly.
>
> Thank you again for your efforts to the review and rebuttal.

---

### Comment · Area_Chair_cYFt · 2024-08-09
**Author-Reviewer discussions (Aug 7 - Aug 13)**

Dear Submission7502 reviewers,

We appreciate your reviews as we move into the Author-Reviewer discussions phase (Aug 7 - Aug 13).
Please read all reviews and author responses carefully.
Please address any remaining questions or concerns with the authors and respond to their rebuttal as needed. Authors need time to respond to your messages, so please post your responses as soon as possible, so there is time for back-and-forth discussion with the authors. At a minimum, please acknowledge that you have read the author rebuttal. Based on the material provided, please adjust your scores and reviews as necessary.

Dear Submission7502 authors,

We appreciate your rebuttal. Please continue the dialogue with the reviewers during this discussion phase.

This message and thread are visible to both reviewers and authors. If you need to respond privately, adjust the "Readers" setting accordingly.

---

### Decision · Program_Chairs · 2024-09-25

**Decision:**

Accept (poster)

**Comment:**

I propose to accept this paper

Strengths:

- Application of LLM in Code Analysis: LLMDFA effectively uses LLMs for code dataflow analysis in bug detection.
- Novel Approach: The three-stage process of source/sink extraction, dataflow summarization, and path feasibility validation
- Promising Results: The approach showed high precision and recall in experiments, outperforming existing baselines and demonstrating potential for real-world applications.
- Customization: LLMDFA's compilation-free design allows it to be adapted to different programming languages with small effort.
- Good Presentation and Structure: The methodology and motivation were clearly articulated, and the empirical evaluation was comprehensive.

Weaknesses:

 - Limited Generalizability: The experiments were only conducted on the Java programming language, raising concerns about the generalizability of the method to other languages.
- Potential for Hallucinations: LLMDFA may still suffer from hallucinations, particularly in dataflow summarization, which could lead to incorrect results.
- Small Dataset: The limited size of the experimental dataset (only 37 programs) could introduce bias, suggesting the need for validation with larger datasets.
- High Cost: The multi-round process required in LLMDFA, particularly in synthesizing scripts for Z3 verification, raised concerns about the cost and efficiency of the approach.

The authors addressed the weaknesses identified by the reviewers as follows:

- Limited Generalizability: During the rebuttal phase, the authors extended LLMDFA to support C/C++ and JavaScript languages and presented additional experimental results, which showed comparable performance to the original Java experiments.

- Potential for Hallucinations: The authors provided detailed statistics on hallucinations and demonstrated that their approach significantly mitigates this issue. They also explained how partially correct path encodings could still filter out false positives, thereby maintaining the effectiveness of the approach.

- Small Dataset: The authors justified the use of the small dataset by pointing to its complexity and relevance in static analysis benchmarks. They also extended their evaluation to other datasets, such as SecBench.js, during the rebuttal phase.

- High Cost: The authors provided a breakdown of the costs associated with each stage of LLMDFA, demonstrating that the financial costs are in line with similar works. They also discussed the potential for further optimization and emphasized the benefits of LLMDFA's customization capabilities.

Overall, the authors' rebuttal successfully addressed most of the reviewers' concerns, leading to an improvement in the reviewers' assessments.